

# Subgrid parameterization of snow distribution at a Mediterranean site using terrestrial photography

Rafael Pimentel[1], Javier Herrero[2], María José Polo[1]

[1]Fluvial Dynamics and Hydrology Research Group. Andalusian Institute for Earth System Research. University of Córdoba.
Campus Rabanales, Edificio Leonardo da Vinci, Área de Ingeniería Hidráulica, 14017, Córdoba, Spain.
[2]Fluvial Dynamics and Hydrology Research Group. Andalusian Institute for Earth System Research. University of Granada.
Edificio CEAMA, Avenida del Mediterráneo s/n, 18006, Granada, Spain.

*Correspondence to*: Rafael Pimentel (rpimentel@uco.es)

**Abstract.** Subgrid variability introduces non-negligible scale effects on the GIS-based representation of snow. This
heterogeneity is even more evident in semiarid regions, where the high variability of the climate produces various
accumulation melting cycles throughout the year and a large spatial heterogeneity of the snow cover. This variability in a
watershed can often be represented by snow depletion curves (DCs). In this study, terrestrial photography (TP) of a cell-
sized area (30x30 m) was used to define local snow DCs at a Mediterranean site. Snow cover fraction (SCF) and snow depth
($h$) values obtained with this technique constituted the two datasets used to define DCs. A flexible sigmoid function was
selected to parameterize snow behaviour on this subgrid scale. It was then fitted to meet five different snow patterns in the
control area: one for the accumulation phase and four for the melting phase in a cycle within the snow season. Each pattern
was successfully associated with the snow conditions and previous evolution. The resulting DCs were able to capture certain
physical features of the snow, which were used in a decision-tree and included in the point snow model formulated by
Herrero et al. (2009). The final performance of this model was tested against field observations recorded over four
hydrological years (2009-2013). The calibration and validation of this DC-snow model was found to have a high level of
accuracy with global RMSE values of 84.2 mm for the average snow depth and 0.18 $m^2$ $m^{-2}$ for the snow cover fraction in the
control area. The use of DCs on the cell scale proposed in this research provided a sound basis for the extension of point
snow models to larger areas by means of a gridded distributed calculation.

## 1 Introduction

Subgrid variability plays a crucial role in GIS-based distributed hydrological modelling. The scale issue introduced when a
point model is applied to a gridded area conditions the accuracy of the processes represented (Blöschl, 1999). This is
especially important in physical modelling because of the non-linearity usually found in natural systems, which does not
allow the assumption of homogeneity within each grid cell.

This is the case of snow models based on energy and water balance (Anderson, 1976; Wigmosta et al., 1994; Luce and
Tarboton, 1996; Herrero et al, 2009). The spatial distribution of snow over mountainous areas can be very heterogeneous. In
cold northern regions, snow can be homogenously distributed as a deep, uniform snowpack, which usually undergoes one





long melting cycle during the year. Nevertheless, in warm mid and low-latitude locations, such as the mountainous areas in Mediterranean-type regions, the changeable climate conditions make the allocation of the snow highly variable and irregular. There may be several accumulation-melting cycles throughout the year, and a wide range of snow-depth states can occur even within small areas. Micro-topography plays an important role in this high spatiotemporal variability, and this effect should be included in gridded representations (Anderton et al. 2004).

Accumulation-snowmelt models tackle this problem from different approaches. Wigmosta et al. (1994) include the effects of local topography and vegetation cover in physical snow modelling in the Distributed Hydrology Soil Vegetation Model (DHSVM). In contrast, the Cold Regions Hydrological Model (CRHM) (Pomeroy et al., 2007), which includes a full range of modules that represent hydrological processes in cold regions, employs a single conception of cascading hydrological response units (HRU). Neither of these models, however, considers the effects of the interaction between micro-topography and snow. This could be a constraint for their application on smaller scales.

Isnobal (Mark and Dozier, 1992) introduces the effect of topography interaction on snow-wind redistribution by determining the formation of drifts and scour zone at the element scale. However, it does not address the effect of low relief on the cell size. Luce et al. (1999) take a snow physically based model and enrich it with the concept of the depletion curve (DC), previously used in runoff prediction based on temperature-index-based melting estimates (Ferguson, 1984; Buttle and McDonnel, 1987). They use DCs for parameterizing subgrid variability in a physical model, which considered the study area as a single model element to reduce the area involved in mass and energy balance as snow regression progresses. These DCs relate a snow state variable, such as snow water equivalent (SWE) or snow depth (h), to the snow cover fraction (SCF) in a selected area. Following this approach, Luce and Tarboton (2004) define a DC in a watershed based on in situ SWE measurements. In addition, Kolbert et al. (2006) used remote sensing information and a Bayesian approach to parameterize a DC on watershed scale. These results show the applicability of this approach to reproduce snow regression in medium to large-sized areas.

Nevertheless, the heterogeneous spatial and temporal snow distribution in semiarid environments makes it difficult to define a single DC for a whole watershed because the evolution of the different snow accumulation-melting cycles that usually occur during the snow season in these regions can differ considerably. A distributed application of DCs could be used to capture this variability, and thus provide a better representation of the physical processes underlying the evolution of snow cover and snow quantity in a given area. This is the approach that was followed in this research study.

A general sigmoid shape is representative of these cycles: (1) an asymptotic trend at the end/beginning of the accumulation-melting processes, respectively, when most of the cell area is covered by snow; (2) a increasing/decreasing trend, corresponding to the period in which snow is falling/melting in the accumulation/melting processes; and (3) a final phase, in which the accumulation/melting begins/finishes and small isolated areas can be found with snow patches.

The basic snow variable needed to define a DC is the snow cover fraction over the area to be scaled. Remote sensing is the most common and powerful extended source to obtain this information in medium and large-sized areas. However, to represent the snow distribution at subgrid scales, higher temporal and spatial resolutions may be required. Terrestrial





photography (TP) of a snow-covered scene is an efficient alternative because its temporal and spatial resolution can be adapted to the scale of the processes driving the snow evolution (Corripio, 2004; Rivera et al., 2008; Farinotti et al., 2010; Pimentel et al., 2012; 2015; Pérez-Palazón et al., 2014). Moreover, TP at this scale also provides snow depth measurements at selected points within the images. This is accomplished by means of coloured rods located in the area that can be clearly
identified and differentiated from the rest of the image.

This study used an innovative approach for incorporating the effects of the spatial variability of the snow distribution at the subgrid scale into snow modelling by means of TP, based on previous work (Pimentel et al., 2015). For this purpose, a 4-yr series of TP images of a 30x30m scene at a snow monitoring site in Sierra Nevada (South Spain) was used to derive DC parameterizations representative of different snow accumulation/melting cycles. The resulting DCs were included in the
snow model developed by Herrero et al. (2009) and Herrero and Polo (2012), and the performance of this DC-model was finally tested against field observations.

## 2 Study site and available data

This study was carried out in the Sierra Nevada Mountains, Southern Spain (37ºN latitude), where the highest altitudes in the Iberian Peninsula can be found (3479 m.a.s.l.). Sierra Nevada is a linear mountain range, which runs 90 km parallel to the
Mediterranean coast. The interaction between the semiarid Mediterranean climate and the alpine conditions in this area results in a highly variable snow regime. The snow usually appears above 2000 m.a.s.l. during winter and spring even though the snowmelt season generally lasts from April to June. The typically mild Mediterranean winters produce several snowmelt cycles before the final melting phase, which distributes the snow in patches over the terrain. Precipitation is heterogeneously distributed over the area because of the steep orography, with a high annual variability (400-1500 mm). The average
temperature during the snow season can range from -5 °C to 5 °C, reaching values as low as -20 °C at certain times in the winter (Pérez-Palazón et al., 2015).

A control area of 900 m$^2$ was selected and sized, according to a grid cell area of 30x30 m. This cell size corresponds to the resolution of Landsat TM scenes, which are used to monitor snow extension over long time periods. This area is located near the weather monitoring station, Refugio Poqueira (Figure 1), at 2500 m.a.s.l. This plot is composed of rocks as well as of
compact, low densely-branched shrubs, mainly *Genista versicolor* and *Festuca indigesta*, which act as insulators between the soil and the snowpack. They constitute the main representative elements of the low relief of Sierra Nevada above 2000 m.a.s.l., where there are only isolated patches of reforested pine trees.

The weather station has been operating since 2004. It generates 5-minute data records pertaining to precipitation, solar radiation, longwave radiation, wind velocity, temperature, air humidity, and atmospheric pressure. In the summer of 2009, an
automatic CC640 Campbell Scientific camera was also installed and programmed to obtain five images per day, every two hours between 8.00 a.m. to 4.00 p.m. of the control area with a resolution of 640 x 504 pixels. This camera is able to capture both the rapid snow melting cycles and the spatial heterogeneity exhibited by the snow cover at the study site in relation to





its micro-topography. Additionally, two snow measuring rods were installed in the photographed area to measure snow depth. Thus, the TP was able to monitor SCF and a representative snowpack depth ($h_{\text{ref}}$) at the study site with a high recording frequency. The SCF and $h_{\text{ref}}$ series in Pimentel et al. (2015), which extend up to the summer of 2013, were used in this work. Tables 1 and 2 show representative climate and snow variables in the control area during the study period.

A digital elevation model (DEM) with a resolution of 0.05x0.05 m was derived from topographic surveys of the control area. This DEM was used to project the TP images and analyse their information.

### 3 Methods

Four consecutive hydrological years (2009-2013) were analysed at the study site. The DCs at subgrid scale were derived from the SCF and average snow depth ($\bar{h}$) values obtained from the TP images during the study period, and were
parameterized by means of flexible sigmoid functions. These DCs were incorporated into the point model of the study site, elaborated by Herrero et al. (2009). This section describes the steps in this process: (i) the definition of DCs; (ii) the way that SCF and $h$ values were obtained from TP; (iii) the point snowmelt-accumulation model; (iv) the inclusion of DCs in the model.

### 3.1 Depletion curves

The $\bar{h}$ over the control area, and the SCF values were selected to express the DCs representative of accumulating-melting cycles. Each cycle corresponds to the time period between the beginning of a snowfall and the end of the associated melting. Each cycle thus has two consecutive phases, accumulation and melting, and for this reason, a different DC was obtained for each phase in every cycle during the study period. In the DC definition, dimensionless $\bar{h}$ ($\bar{h}^* = \bar{h}/\bar{h}_{\text{max}}$ [x-axis]) and SCF ($SCF^* = SCF/SCF_{\text{max}}$ [y-axis]) values scaled based on the maximum average snow depth ($\bar{h}_{\text{max}}$) and the maximum SCF
($SCF_{\text{max}}$), of each cycle respectively, were used to minimize the variability throughout the cycles in the study period. Both variables were obtained from the TP series recorded at the study area (see Sect. 3.2). The flexible sigmoid function given by Eq. (1) was used as the DC parameterization. This function, which was adapted from the proposed by Ying et al. (2003) to characterize crop growth, provides a flexible description of asymmetrical sigmoid patterns, numerical stability of parameters in statistical estimation, prediction of a zero value at the origin of the coordinates, and easy truncation to represent
asymptotic behaviour.

$$SCF_i^* = \begin{cases} SCF_{i_{\max}}^* \left(1 + \dfrac{\bar{h}_{i_e}^* - \bar{h}_i^*}{\bar{h}_{i_e}^* - \bar{h}_{i_m}^*}\right) \left(\dfrac{\bar{h}_i^*}{\bar{h}_{i_e}^*}\right)^{\frac{\bar{h}_{i_e}^*}{\bar{h}_{i_e}^* - \bar{h}_{i_m}^*}} & if\ 0 \le \bar{h}_i^* < \bar{h}_{i_e}^* \\ SCF_{i_{\max}}^* & if\ \bar{h}_i^* \ge \bar{h}_{i_e}^* \end{cases} \tag{1}$$

This function has three parameters, $SCF_{i_{max}}^*$, $\bar{h}_{i_e}^*$, and $\bar{h}_{i_m}^*$, which are, respectively, the maximum value of $SCF^*$ for a given cycle, the dimensionless snow depth when the melting begins, and the dimensionless snow depth when the maximum melt rate is reached during a cycle. The subindex $i$ represents each of the cycles during the study period.





The time evolution of the SCF during the study period was analysed to define the set of accumulation/melting cycles. The accumulation phase exhibited low variability whereas the melting phase of the cycles showed considerable variation. Consequently, a single sigmoid function fits all the accumulation phases in the selected cycles, whereas different curves were fitted for each snowmelt phase. The fitting procedure was iterative to cluster the DCs after cycles with a similar regime. DCs with close fitted values (10% difference) were clustered, and then a second fit was performed for the whole data set, and so on. Finally, five parameterizations were selected: one for the accumulation phase and four for the different types of melting phase.

### 3.2 Terrestrial photography analysis: SCF and snow depth measurements

The SCF and $\bar{h}$ series used in this work were obtained by following the methodology proposed by Pimentel et al. (2012; 2015) and Pérez-Palazón et al. (2014), which consists of a two-step method involving georeferencing and snow detection. Firstly, georeference provided the photo with spatial references in order to identify a function that related the 3D-point in the DEM to the associated 2D-pixel in the image. For this purpose, standard automatic computer vision algorithms were applied (Fiume et al. 1989; Foley et al. 1990). Secondly, the snow detection was performed with a clustering algorithm, namely, a K-means algorithm (MacQueen, 1967), which classified pixels into two clusters: snow covered and non-covered. Following this pixel classification, the SCF in each image was easily calculated by the sum of pixels in each cluster (Pimentel et al., 2012).

The average snow depth, $\bar{h}$, was obtained in each image using the rods installed in the plot. They were painted bright red, which made them easy to distinguish from the other elements in the photo. The same clustering algorithm was applied to determine rod and no-rod pixels. A previously defined linear equation was applied to the thus identified rod-pixels to transform them into a reference snow-depth value ($h_{ref}$). The associated $\bar{h}$ value for the whole control area was estimated for each image by assuming the uniform horizontal accumulation of snow (Luce et al., 1999) and using the empirical relationship (Eq. 2) between the $h_{ref}$ value in the rod and the value of $\bar{h}$ from the analysis of the topography of the control area (Figure 2). A piecewise function, composed of a cubic expression for a reference snow depth below the influence height of the micro-topography, 0.6 m (calculated as the 99th percentile of the micro-topography distribution in the control area), and a linear relationship for reference snow depth values higher than this threshold were obtained, where both $\bar{h}$ and $h_{ref}$ are expressed in meters Eq. (2).

$$\bar{h} = \begin{cases} -0.2764 \cdot h_{ref}^3 + 0.8066 \cdot h_{ref}^2 + 0.2608 \cdot h_{ref} & if \ h_{ref} \leq 0.6 \ m \\ h_{ref} - 0.2129 & if \ h_{ref} > 0.6 \ m \end{cases} \tag{2}$$





### 3.3 Snow modelling

### 3.3.1 Point model

The snowmelt-accumulation model for a Mediterranean site, developed by Herrero et al. (2009) is a point physical model based on a mass and energy balance. The model assumes a uniform horizontal snow cover surface distributed in one vertical

layer. This snow column per unit area defines the control volume, which has the atmosphere as an upper boundary and the ground as the lower one. The lateral mass and energy fluxes between adjacent snow columns are regarded as null. The energy and mass balances are given by Eqs. (3-4):

$$\frac{dSWE}{dt} = R - E + W - M \tag{3}$$

$$\frac{d(SWE \cdot u)}{dt} = \frac{dU}{dt} = K + L + H + G + R \cdot u_R - E \cdot u_E + W \cdot u_W - M \cdot u_M \tag{4}$$

where $SWE$ is the water mass in the snow column, and $u$ is the internal energy per unit of mass ($U$ for total internal energy). In the mass balance, $P$ defines the precipitation flux; $E$ is the flux of water vapour diffusion (evaporation/condensation); $W$ represents the mass transport flux due to wind; and $M$ is the melting water flux. On the other hand, regarding the energy fluxes, $K$ is the solar or short-wave radiation; $L$, the thermal or long-wave radiation; $H$, the flux of sensible heat exchange with the atmosphere; $G$, the flux of heat exchange with the soil; and $u_R, u_E, u_W,$ and $u_M$ are the advective heat flux terms

associated with each of the mass fluxes in Eq. (3), respectively.

In the mass balance Eq. (3), $W$ was disregarded because of the rapid snow metamorphosis, which compacts the snow and reduces its mobility. $G$ was not considered in the energy balance equation Eq. (4) since it is regarded as a secondary term per se in this balance (Kuusisto, 1986). The calculation of the short-wave radiation ($K$) was based on the measured downwelling short-radiation flux and the albedo (Pimentel et al., 2015; Aguilar et al., 2010). The calculation of the longwave radiation

was based on the formulation for atmospheric longwave emissivity. developed in the Sierra Nevada area by Herrero and Polo. (2012). Finally, the $H$ term was modelled as a diffusion process (Dingman, 2002). A detailed description of the formulation of each term can be found in Herrero et al. (2009).

### 3.3.2 Incorporation of depletion curves in the point model

The point model described in the previous section was expanded to perform calculations for a 30x30 m cell by including the

25 DCs obtained from TP. In this way, SCF became a new state variable in the snow model. In time step $t_1$, when a snowfall event begins, the mass and energy balances are solved for the whole cell, and the snow state variables are calculated per unit of area. SCF at this time ($SCF_1$) is estimated from the $\bar{h}$ calculated at this time ($\bar{h}_1$) and by using the associated DC. In the next time step ($t_2$), $SCF_1$ is used as a reduction factor of the area affected by the mass and energy balance. This iterative process is repeated in the model whenever snow is still present in the cell.





Three hydrological years were used for calibration: 2009-2010, a wet year; 2010-2011, a very wet year; and 2011-2012, a dry year. The stability correction factors for turbulent transfer ($\phi_M$, $\phi_V$ and $\phi_W$) (Cline, 1999), the sensible-heat transfer coefficient in windless conditions ($K_{H_0}$), and the value of snow surface roughness ($z_0$) are the calibration parameters in the snow model by Herrero et al. (2009). A fourth year of data (2012-2013) was used to validate the previous results. The mean

error (ME), mean-absolute error (MAE), and root-mean-squared error (RMSE) were employed in the calibration and validation process as an objective function to be minimized and tested in the validation step with SCF and $\bar{h}$ as test variables.

## 4. Results

### 4.1 Terrestrial photography measurements

Figure 3 shows some representative examples of the SCF maps obtained from the TP georeferencing and snow detection processes. As can be observed, the detection algorithm was able to capture snow presence even under adverse atmospheric conditions such as foggy and extremely cloudy days.

The temporal evolution of $\bar{h}$ and SCF, both measured from the TP images, is represented in Figure 4. A high variability is observed in the accumulation-melting cycles that occurred during the four years of the study period. The number of cycles

and their duration varied considerably over the years, with a mean number of $18 \pm 5$ cycles per year and a mean duration of $49\pm14$ and $108\pm18$ days for the accumulation and melting phases of each cycle, respectively. Furthermore, the mean daily snow depth fluctuated greatly from year to year, ranging from 0.12 m in the driest year (2011-2012) to 0.56 m in the wettest (2010-2011). There was a maximum average snow depth, $\bar{h}$, of 1.19 m in the most humid year. As for the SCF, each year the control area was completely covered for at least for one day, with mean daily SCF values ranging from 0.50 $m^2m^{-2}$ to 0.88

$m^2m^{-2}$ (Table 2).

In order to analyse subgrid variability, 16 of the 53 cycles during the three calibration years (2009-2010, 2010-2011, 2011-2012) were finally selected. Those cycles that either had a short duration of less than 5 days, or took place when the study area was completely covered by snow, were not considered in the study since they did not provide a significant amount of information. The events finally selected are those marked with a red circle in Figure 4.

### 4.2 The formulation of depletion curve and snow processes

The DC given by Eq. (1) was fitted to each of the cycles in Figure 4, as explained in Sect. 3. Since the duration of the accumulation phase showed a low deviation when compared to that of the melting phase in each cycle, the data of this phase were jointly considered in the procedure. In contrast, the melting phases were individually treated for each event. Table 3 shows the fitted values of the parameters associated with the melting phase of each event. As explained in Sect. 3.1, these





individually fitted parameters during the melting phase were used to cluster the melting phases and identify common patterns within the cycles. The final results clustered the melting phases in four groups (identified as curves 1-4 in Table 3). The associated values of their fitted parameters are shown in Table 4 together with those corresponding to the accumulation phase (curve 0).

Regarding the five DCs given by the values in Table 4, their shape can be associated with the description of the evolution of the snowpack within the control area (Figure 5). More specifically, Curve 0, which describes the accumulation phase of the selected cycles, produces a very fast initial accumulation that covers almost half of the maximum snow area during the accumulation phase. This rapid accumulation is followed by a somewhat slower increase, which finishes with a dimensionless snow depth value of 0.7, a maximum threshold beyond which the area is completely covered. As for the

melting phases, Curve 1 is representative of cycles with a large amount of snow (a high $\bar{h}^*$ value) resulting from a long accumulation phase, and it is associated with a very compact state of the snow with a high level of metamorphism.

Curve 2 also represents melting phases with a large quantity of snow, with high initial values of snow depth. In this case, they are preceded by a short, non-persistent accumulation phase. Moreover, since melting occurs directly after the accumulation process ends, this snow is only slightly compacted. Finally, both Curve 3 and Curve 4 represent snowmelt

cycles with significantly lower snow depth values. In these cases, the micro-topography is very likely the direct and main driver of the melting process from the beginning of this phase. In fact, the length of the snow season is the main difference between both curves, autumn or winter cycle (Curve 3) in comparison to the spring cycle (Curve 4). In the cold months, the snowmelt rate has a fast initial value followed by a slowdown in its final stage. In contrast, in the warmer spring months, the snow decay is faster and approximately constant throughout the melting phase.

This difference in dynamics is described by the fitted parameters values in the associated DCs (Table 4). More specifically, $\bar{h}_e^*$ defines the beginning of the melting phase in each cycle. It is lower in Curve 1 than in Curve 2 (0.823 vs 0.905) and reflects the effect of snow consolidation, which delays the beginning of the melting phase. Furthermore, its value is 1 in Curves 3 and 4, as corresponds to cycles in which the melting phase immediately occurs after the accumulation maximum. Regarding $\bar{h}_m^*$, which determines the moment with maximum melting rates, it drops to zero in Curves 1 and 4, which are

associated with quasi-stationary melting rates during this phase. In contrast, melting phases represented by Curve 2 ($\bar{h}_m^* = 0.488$) and Curve 3 ($\bar{h}_m^* = 0.727$) exhibit two different rates: a fast value and a slow value, associated with the initial and final melting stages, respectively.

Based on the previous data, a decision-tree was defined to incorporate the corresponding depletion curve given by the fitted parameters in Table 4 into the snowmelt-accumulation point model. Figure 6 shows this decision tree, which answers this

sequence of questions: (1) the duration of the snow (number of days greater or lesser than 30) in the control area when the melting phase begins to discriminate the effects on the compaction of the snowpack; (2) maximum snowpack depth during a given cycle (a threshold value of 0.60 m was identified as representative of the topography of the control area); and (3) month when the cycle occurs to discriminate between autumn-winter and spring.





### 4.3 Calibration and Validation

Once the decision-tree was implemented into the snow point accumulation-melting model with the DCs given by the fitted parameter values in Eq. (1) (Table 4), different simulations were performed to optimize the performance of the model in regard to the daily $\bar{h}$ and SCF values during the calibration period. Table 5 shows selected simulations for both variables
with different calibration hypotheses, and the comparative statistics for each of the variables.

The simulations reflect the expected sensitivity of the model to the different calibration parameters. As can be observed, the use of null values of the $\phi$ factors together with high values of both $K_{H_0}$ and $z_0$ accelerate the melting stage (reflected in the positive values of ME in Simulations 1, 2 and 3 in Table 5). Furthermore, this sensitivity is more enhanced for $\bar{h}$ estimation than for SCF estimation (e.g. the reduction of the RMSE value from 214.9 mm to 126.7 mm between Simulations 1 and 3).

In the same way, low values of these two parameters also delay the extinction of the snow and result in more mismatching between observed and simulated values (e.g., Simulation 5). The optimal calibration values selected for the snow model modified by DCs were those for Simulation 7 (Table 5): (a) null values of $\phi$ factors, as proposed in other studies (e.g. Tarboton and Luce, 1996); (b) $K_{H_0} = 1\,\mathrm{W\,m^{-2}\,K}$, which is a value lower than what Herrero et al. (2009) found from point snow depth measurements at the study site, but closer to those proposed by Jordan (1991) and Jordan et al. (1999) in

previous work; and (c) $z_0 = 2.5$ mm, which lies within the range of 0.0001-0.01 m, as proposed by various authors (i.e. Jordan, 1991; Marks and Dozier, 1992; Tarboton and Luce, 1996; Marks *et al.*, 2008; Andreadis *et al.*, 2009).

Figures 7 and 8 show the values of $\bar{h}$ and SCF, respectively, during both the calibration and validation periods, along with the values simulated with the optimal calibration parameters. The simulation of SCF (Figure 7) resulted in a high level of overall accuracy, with ME=0.04 mm, MAE=0.079 mm, and RMSE= 0.180 mm. All the snowmelt cycles are represented

though the simulated values generally overestimate the dataset measured during short snowmelt cycles, i.e. at the beginning of the first year or during the winter of the third year. In contrast, the model tends to underestimate the SCF during snow-persistent periods. The simulation of the average snow depth (Figure 8) also satisfactorily reproduced snow behaviour during the calibration period with a global RMSE=84.2 mm. This accuracy, however, is lower in the first year, from the beginning of the winter, in which measured and simulated values show a clear mismatch even though their trends follow the same

pattern. The other two years in the calibration period are accurately represented, and the different snowmelt cycles are adequately reproduced. This includes both the intense rates at the beginning and the end of the annual snow season as well as the long periods with a persistence of snow.

Both Figures 7 and 8 also include the results of the model in contrast to the observations during the fourth year of data (2012-2013) in the form of a validation period (graph b in both figures) of the parameters calibrated in Simulation 7 (Table

5). Table 6 shows the results and the statistics of this validation for both variables, $\bar{h}$ and SCF, respectively. The results generally reproduced statistics similar to those achieved for the calibration period with the same behaviour observed during calibration: a general overestimation of SCF during short cycles and a mismatch in the simulated snow depth values for certain states.





## 5 Discussion

The 5 DCs explain the accumulation-melting snow behaviour with the four melting patterns observed during the study period: (a) cycles with a large amount of consolidated snow; (b) cycles with a large amount of recent snow; (c) cycles with a small amount of snow during autumn or winter; and (d) cycles with a small amount of snow during spring. Furthermore, they

succeeded in parameterizing the spatial distribution of the snow patches with the same type of curve. To illustrate this, Figure 9 shows three images during each of the three cycles represented by the same curve (Curve 2). They correspond to three SCF states during the cycle (90%, 50%, and 20% of the snow coverage in the control area).

A different snow distribution can be observed in images from different dates but with the same SCF value. These differences stem from the driving atmospheric conditions during the accumulation phase and the beginning of the melting stage, and

their interaction with the micro-topography. For example, in the images for 2011-2012, Cycle 13, the effect that wind produced on the snow distribution can be clearly observed, with small accumulation areas close to the largest rocks.

However, all of these distributions can be accurately represented with the same curve. Hence, this parameterization captures the subgrid variability without further need to physically model the process at this scale, such as the interaction between wind and micro-topography, among others. Similar effects on the snow distribution can be observed in the sequence of states

associated with the other curves. This means that this parameterization can be extended to other control areas with similar micro-topography and conditions.

The inclusion of this 5-curve set of DCs improved the performance of the snow model obtained in previous work, which tested both a general DC formulation and data assimilation by using Ensemble Kalman Filter methods (Pimentel et al., 2015). Therefore, this semi-empirical approximation showed itself to be powerful and at the same time, less computationally

costly than those data assimilation techniques.

In spite of the general accuracy of these results, some mismatches occurred in certain cycles during the study period, identified as A to H and I to K, respectively, for the calibration and validation periods in Figure 8. They can be classified according to the following error sources:

1) An incorrect determination of the precipitation fraction occurring in the form of rain or snow (blue circles in Figure 8). In

this regard, the model considers precipitation as snow whenever the wet-bulb temperature is under 0ºC, an assumption that generally leads to a correct representation of snowfall events. However, in certain events, this threshold overestimates the amount of snowfall, such as in cycles A and E at the beginning of winter (Figure 8), and in all likelihood, it also underestimates this amount in the spring cycles (D, G, H and K in Figure 8).

2) The apparently insufficient representation of the effects of the rain-over-snow events (green circles in Figure 8) by the

model.

3) The impossibility of capturing the effects of the blowing snow associated with high gusts of wind (red circles in Figure 8). The model does not incorporate snow transport by the wind (section 3.2). Despite the fact that DCs can capture this wind





redistribution within the subgrid scale, at least to a certain extent, they cannot reproduce the net snow transfer from adjacent areas.

TP images facilitate the analysis of the previously mentioned error sources of error since the three factors can be identified from the original images during a given accumulation-melting cycle. This has the additional advantage of implementing this

technique in snow monitoring networks, especially in highly variable conditions, such as those characterizing the climate and snow regime of Mediterranean regions.

## 6 Conclusions

This study analysed the subgrid variability of the snow distribution in a Mediterranean region and formulated a parametric approach that includes these scale effects in the physical modelling of snow by means of depletion curves associated with

10 snowmelt patterns. The use of terrestrial photography permitted a continuous monitoring of the snow distribution that can be easily adapted to both the spatial and temporal small-scale effects of the physical processes governing the accumulation-melting cycles. In this work, TP economically monitored the evolution of SCF and $\bar{h}$ over a 30x30m control area in order to study the subgrid variability of the snow, which cannot be captured by other more conventional remote sensors.

The TP-information provided the data for the definition of depletion curves corresponding to different patterns of snow

accumulation-melting cycles within the annual snow season, which is a usual feature of Mediterranean environments. The five groups of DCs succeeded in capturing the subgrid variability of different drivers at this scale, which have a direct effect on snow distribution, mainly the interaction between wind and microtopography. The results show the importance of including different DC patterns instead of a single one to accurately represent snow behaviour during the different cycles within a snow season. Moreover, greater variability was found in the melting phase patterns in the cycles than in those of the

accumulation phase. All of the patterns of the accumulation phase were almost the same, whereas up to four patterns were found for the melting phase.

These four DCs were found to be associated with the age of the snow, and the dominant atmospheric conditions during the melting. These data were used to derive a decision tree, which was included in the snow model (Herrero et al. 2009) used in the study. The tree had the three following decision indicators: (i) the number of antecedent days with snow; (ii) the amount

of accumulated snow previous to melting; and (iii) the month of the snow season.

The final optimal calibration of the DC-snow model improved the results previously obtained at the study site by Pimentel et al. (2015). These include assimilation techniques in the modelling, especially for the simulated snow depth values, with a global error of less than 84.2 mm, and a similarly satisfactory representation of SCF values with an error of less than 0.18 $m^2m^{-2}$.

Despite this improvement, the performance of the DC-snow model was not always satisfactory since certain cycles were not explained by the modelling. However, the information provided by the TP images permitted the analysis of potential error sources as well as the identification of additional drivers of the subgrid scale effects, such as the occasionally incorrect determination by the model of the precipitation fraction in the form of rain or snow, the net transport of snow from adjacent





areas by strong gusts of wind and rain, and a poor representation of the rain-over-snow effects. Further work is currently being carried out to improve the representation of these modelling conditions. TP is a valuable data source that complements standard weather observations (i.e. overestimation of rain/snow measurements) and contributes toward a better understanding of the snow behaviour under certain drivers (i.e. wind advection).

5    The results confirm that the use of DCs on a cell scale, as proposed in this work, provides a solid foundation for the extension of point snow models to larger areas by means of a gridded distributed calculation.

**Acknowledgements**

This work was funded by the Spanish Ministry of Economy and Competitiveness - MINECO (Research Project CGL 2011-25632, "Snow dynamics in Mediterranean regions and its modelling at different scales. Implication for water management"

10   and Research Project CGL2014-58508R, "Global monitoring system for snow areas in Mediterranean regions: trends analysis and implications for water resource management in Sierra Nevada"). Moreover, this research was partially developed within the framework of the Panta Rhei Research Initiative of the International Association of Hydrological Science (IAHS) in the Working Group on Water and Energy Fluxes in a Changing Environment.





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



# List of tables

**Table 1: Statistical descriptors of selected meteorological variables at the Refugio Poqueira weather station during the study period (2009-2013).**

|  | Mean | Max | Min |
|---|---|---|---|
| Annual precipitation (mm) | 1072 | 1460 | 537 |
| Annual snowfall fraction (% of the annual precipitation) | 65 | 83 | 45 |
| Daily temperature during the year (ºC) | 6.7 | 24.7 | -15 |
| Daily temperature during the snow season (ºC) | 1.3 | 16.3 | -15 |
| Daily solar radiation (J m$^{-2}$) | $20.5 \cdot 10^6$ | $35.8 \cdot 10^6$ | $0.5 \cdot 10^6$ |
| Daily wind speed (m s$^{-1}$) | 3.6 | 13.5 | 0.1 |





**Table 2: Statistical descriptors of the SCF and $\bar{h}$ datasets measured by using TP in the control area (Pimentel et al., 2015) for each year and during the whole study period (2009-2013).**

| | 2009-2010 | 2010-2011 | 2011-2012 | 2012-2013 | Study period |
|---|---|---|---|---|---|
| mean SCF ($m^2\ m^{-2}$) | 0.8 | 0.88 | 0.5 | 0.75 | 0.75 |
| maximum SCF ($m^2\ m^{-2}$) | 1 | 1 | 1 | 1 | 1 |
| mean $\bar{h}$ (m) | 0.30 | 0.56 | 0.12 | 0.31 | 0.1 |
| maximum $\bar{h}$ (m) | 1.04 | 1.19 | 0.41 | 0.88 | 1.19 |
| number of accumulation-melting cycles | 24 | 17 | 12 | 18 | 71 |
| mean duration of the accumulation phase in each cycle (days) | 62 | 55 | 29 | 50 | 196 |
| mean duration of the melting phase in each cycle (days) | 119 | 127 | 87 | 99 | 432 |





**Table 3. Fitted values of the parameters in Eq. 1 ( $h_e^*$ and $h_m^*$ ) and the associated determination coefficient ($R^2$) for the melting phase of each selected cycle during the calibration period. The last column to the right shows the groups in which each DC was finally clustered (identified as Curves 1 to 4).**

| Year | Cycle | $h_e^*$ | $h_m^*$ | $R^2$ | Type |
|---|---|---|---|---|---|
| 2009-2010 | 9 | 0.80 | 0.14 | 0.85 | Curve 1 |
| 2009-2010 | 11 | 0.75 | 0.15 | 0.90 | Curve 1 |
| 2009-2010 | 13 | 0.85 | 0.10 | 0.88 | Curve 1 |
| 2009-2010 | 22 | 0.78 | 0.13 | 0.94 | Curve 1 |
| 2010-2011 | 2 | 0.80 | 0.40 | 0.97 | Curve 2 |
| 2010-2011 | 15 | 0.76 | 0.16 | 0.97 | Curve 1 |
| 2010-2011 | 16 | 1.10 | 0.05 | 0.97 | Curve 4 |
| 2010-2011 | 17 | 1.15 | 0.13 | 0.90 | Curve 4 |
| 2011-2012 | 2 | 0.91 | 0.56 | 0.97 | Curve 2 |
| 2011-2012 | 4 | 0.99 | 0.77 | 0.93 | Curve 3 |
| 2011-2012 | 5 | 1.00 | 0.76 | 0.97 | Curve 3 |
| 2011-2012 | 6 | 1.00 | 0.65 | 0.97 | Curve 3 |
| 2011-2012 | 7 | 1.05 | 0.80 | 0.89 | Curve 3 |
| 2011-2012 | 8 | 1.00 | 0.65 | 0.96 | Curve 3 |
| 2011-2012 | 9 | 1.01 | 0.45 | 0.95 | Curve 4 |
| 2011-2012 | 11 | 0.85 | 0.38 | 0.97 | Curve 2 |




**Table 4. Fitted values of the parameters in Eq. 1 ($h_e^*$ and $h_m^*$) and the associated determination coefficient ($R^2$) for both the accumulation pattern (Curve 0) and melting phase patterns (Curves 1 to 4) identified within the selected cycles during the calibration period.**

| Curve | $h_e^*$ | $h_m^*$ | $R^2$ |
|---|---|---|---|
| Curve 0 | 0.707 | 0.000 | 0.85 |
| Curve 1 | 0.759 | 0.000 | 0.91 |
| Curve 2 | 0.861 | 0.264 | 0.94 |
| Curve 3 | 1.000 | 0.617 | 0.85 |
| Curve 4 | 1.000 | 0.157 | 0.98 |





**Table 5. Performance of the snow model during the calibration period, including the DCs in Table 4 and Figure 6: calibration values used in the selected simulation trials along with the associated error indicators for both daily $\bar{h}$ and SCF.**

| Average snow depth, $\bar{h}$ | | | | | | |
|---|---|---|---|---|---|---|
| Simulation | $(\phi_M, \phi_V, \phi_W)$ | $K_{H_0}$ (W m$^{-2}$ K$^{-1}$) | $z_0$ (mm) | ME (mm) | MAE(mm) | RMSE(mm) |
| 1 | None | 6 | 25 | 85.1 | 93.2 | 214.855 |
| 2 | Cline | 6 | 25 | 65.0 | 74.7 | 171.2 |
| 3 | Cline | 1 | 25 | 43.1 | 57.2 | 126.7 |
| 4 | Cline | 1 | 5 | -19.4 | 50.4 | 98.1 |
| 5 | Cline | 1 | 1 | -77.0 | 88.3 | 156.8 |
| 6 | None | 1 | 1 | -28.3 | 50.2 | 98.7 |
| **7** | **None** | **1** | **2.5** | **-0.7** | **39.6** | **84.2** |
| 8 | None | 2 | 2.5 | 5.0 | 39.3 | 85.7 |
| Snow cover fraction, SCF | | | | | | |
| Simulation | $(\phi_M, \phi_V, \phi_W)$ | $K_{H_0}$ (W m$^{-2}$ K$^{-1}$) | $z_0$ (mm) | ME (m$^2$ m$^{-2}$) | MAE(m$^2$ m$^{-2}$) | RMSE(m$^2$ m$^{-2}$) |
| 1 | None | 6 | 25 | 0.069 | 0.109 | 0.243 |
| 2 | Cline | 6 | 25 | 0.028 | 0.091 | 0.207 |
| 3 | Cline | 1 | 25 | 0.002 | 0.093 | 0.205 |
| 4 | Cline | 1 | 5 | -0.070 | 0.102 | 0.221 |
| 5 | Cline | 1 | 1 | -0.118 | 0.135 | 0.259 |
| 6 | None | 1 | 1 | -0.066 | 0.091 | 0.201 |
| **7** | **None** | **1** | **2.5** | **-0.040** | **0.079** | **0.180** |
| 8 | None | 2 | 2.5 | -0.033 | 0.077 | 0.177 |





**Table 6. Values of the error indicators for both snow depth and SCF variables obtained for the DC-snow model during the validation period with the calibration parameters in Simulation 7 (Table 5).**

| Variable | ME | MAE | RMSE |
|---|---|---|---|
| $\overline{h}$ (mm) | -5.110 | 66.841 | 105.8 |
| SCF ($m^2\ m^{-2}$) | -0.112 | 0.134 | 0.219 |



**Figure 1. Location of the study site at the Sierra Nevada Mountains, Spain (top), and DEM of the control area located near the Refugio Poqueira (RP) weather station (bottom). The black dot indicates the location of the weather station and the black solid line, the area covered by the images obtained from terrestrial photography.**





**Figure 2: A simplified 2D representation of the relationship between $h_{\mathrm{ref}}$ and $\overline{h}$ in each TP image of the control area.**

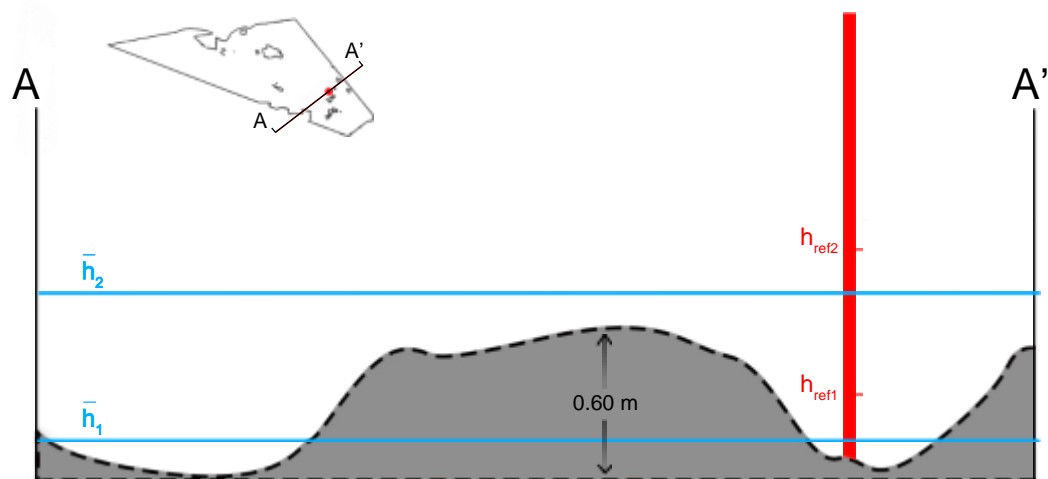




**Figure 3. Selected examples of the final SCF maps (0.05x0.05m) obtained from the terrestrial photography treatment procedure at the study site for certain representative states during the snow season: (a) original images for each date (30x30 m); (b) georeferenced images; (c) snow masks obtained from the georeferenced image by using a K-mean algorithm (snow presence in red).**

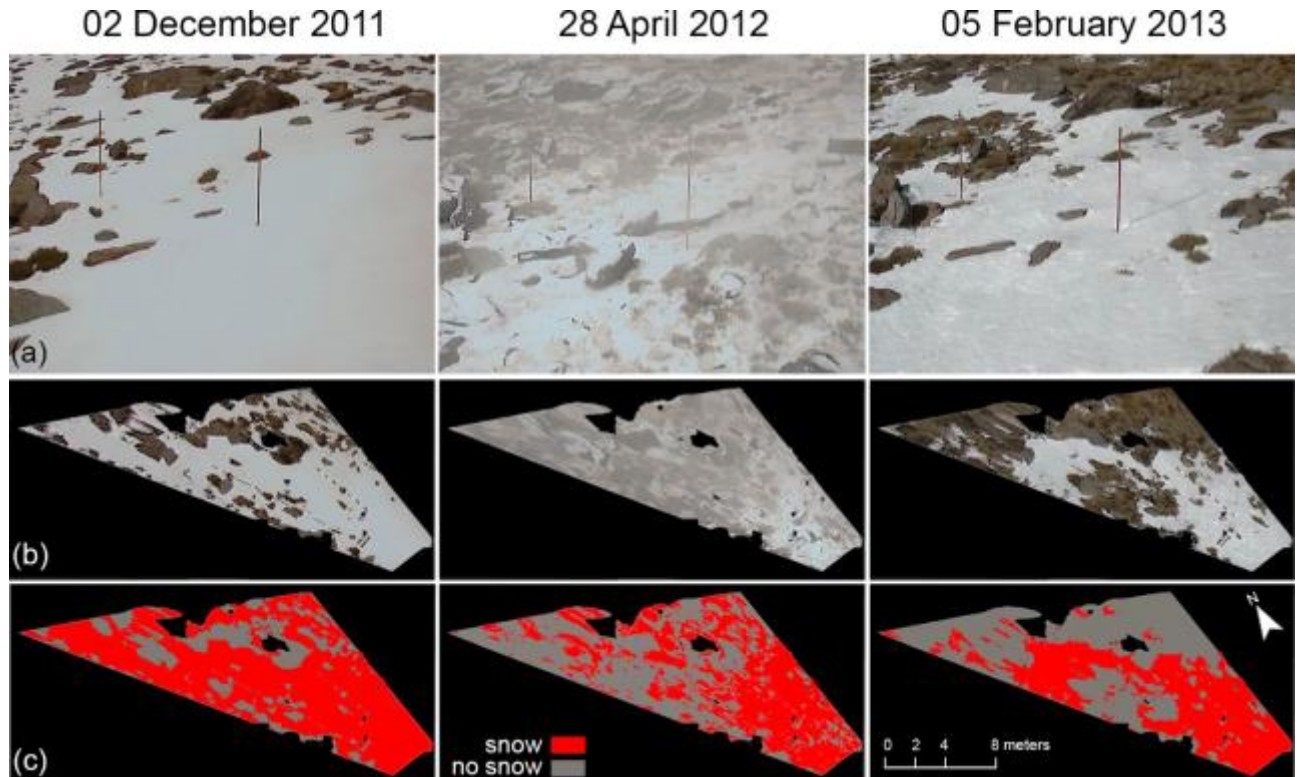





**Figure 4.** Temporal evolution of both the daily snow cover fraction, SCF (gray bars) and average snow depth, $\bar{h}$ (black line), obtained from TP at the control area during the 4-yr study period. The accumulation-melting events are numbered for each year. Red circles indicate the events selected for the subgrid variability analysis during the 3-yr calibration period.

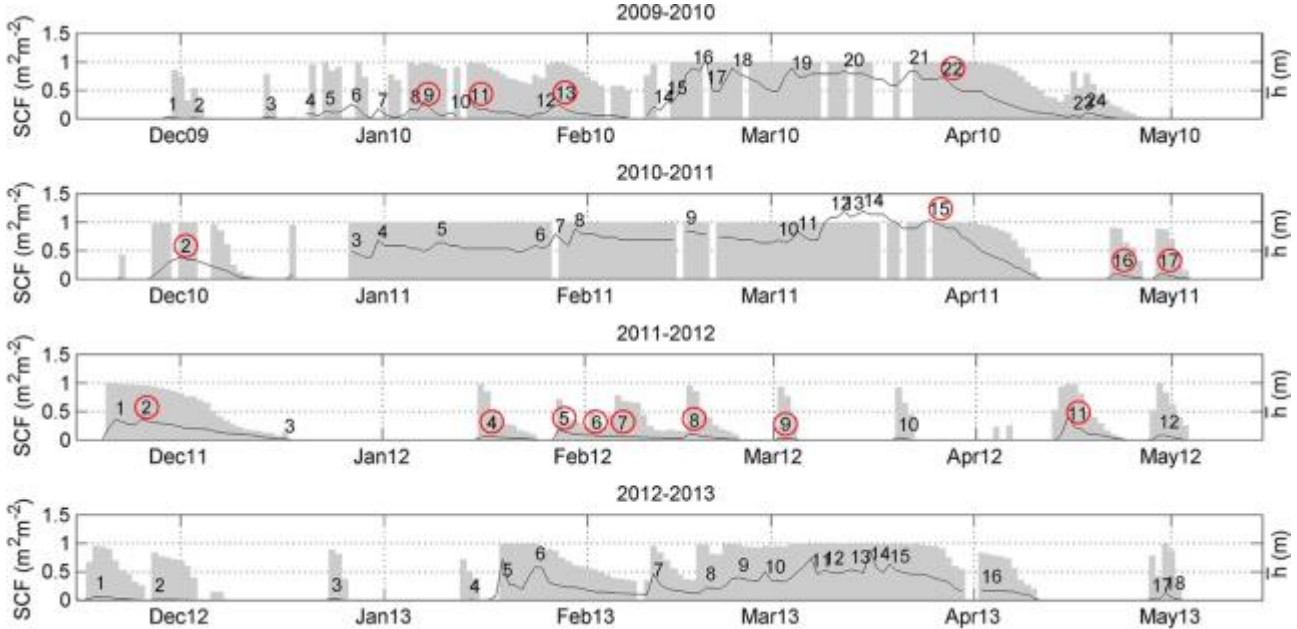





**Figure 5. (a) Accumulation-melting cycles used in the study. These include selected cycles during each year of the calibration period (2009-2012) with their classification in the depletion curves (DCs) in Table 5. (b) Also included are the following DC patterns: Curve 0, accumulation phase for all the cycles; Curve 1, melting phase of cycles with initial high snow depth, following a long accumulation stage; Curve 2, melting phase of cycles with initial high snow depth, following a short accumulation stage; Curve 3, melting phase of cycles with low snow depth at the beginning of the annual snow season (autumn-winter); and Curve 4, melting phase of cycles with low snow depth at the end of the annual snow season (spring).**

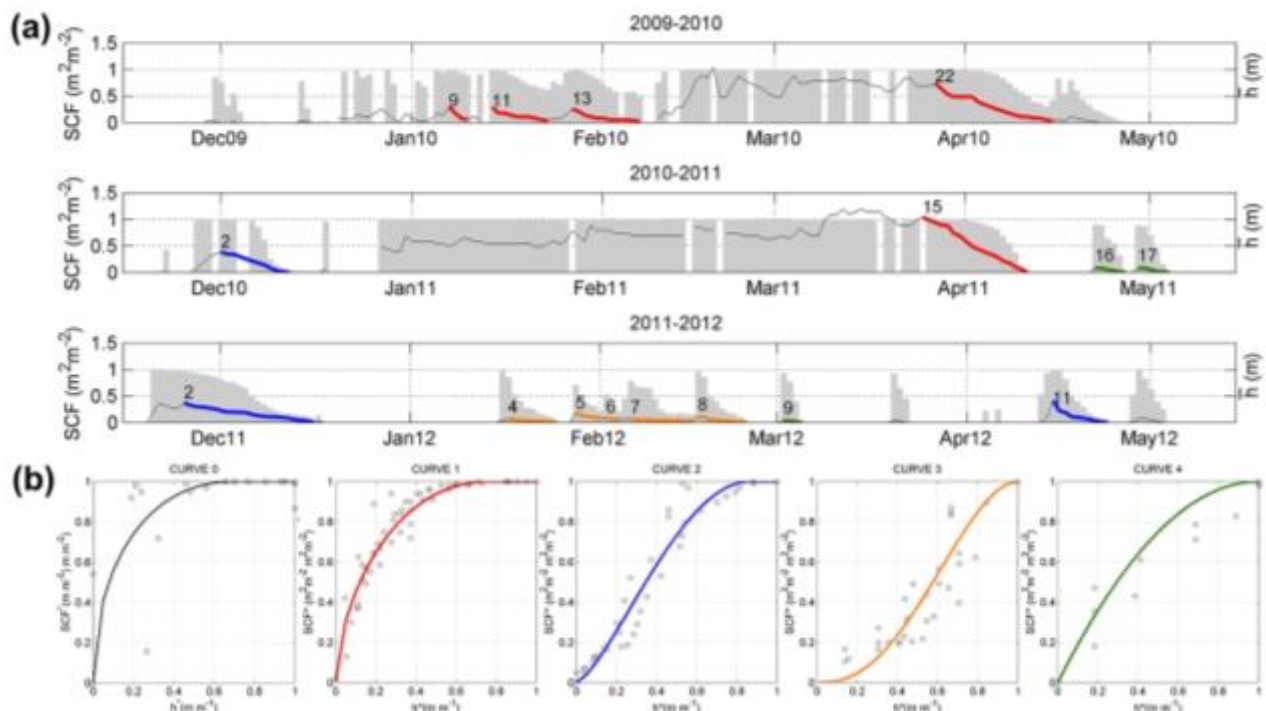




**Figure 6. Decision tree included in the snow model extension for a control area of 30x30 m to select the depletion curve associated with each accumulation-melting cycle during the snow season.**

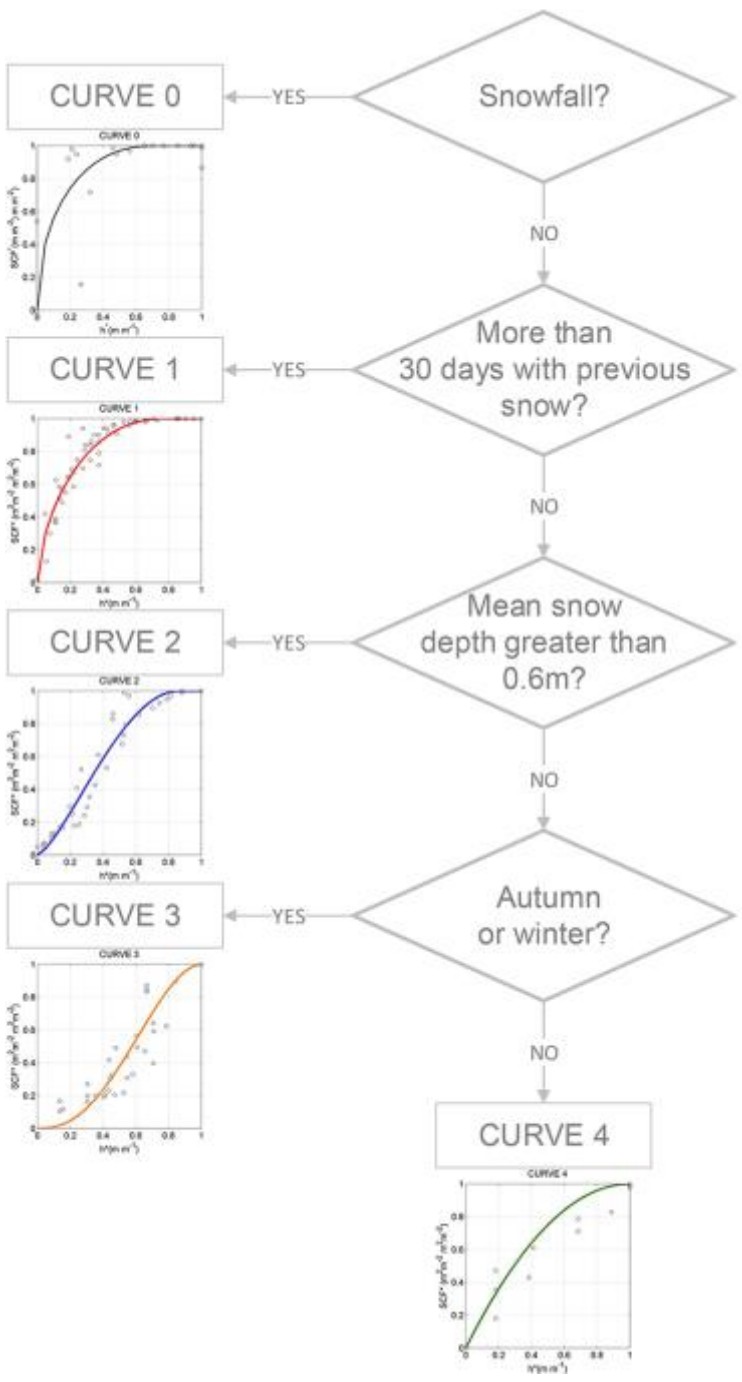




**Figure 7. Temporal evolution of the measured and simulated daily SCF values, based on the calibration parameters in Simulation 7 (Table 5) during (a) the calibration period and (b) the validation period. Dispersion graphs for each year are also provided on the right.**

(a)

(b)





**Figure 8.** Temporal evolution of the measured and simulated daily $\bar{h}$ values from the calibration parameters in Simulation 7 (Table 5) during the (a) calibration period and (b) the validation period. Dispersion graphs for each year are also provided on the right. Mismatches are identified from A to K, and coloured circles are associated with the potential error sources in the modelling.





**Figure 9. Selected examples of snow distribution patterns during the snowmelt phase of three different cycles represented by the same depletion curve (Curve 2).**