# Peer review of "Subgrid parameterization of snow distribution at a Mediterranean site using terrestrial photography"

_Hydrology and Earth System Sciences, 2016_

## Referee Comment (RC1) · Anonymous Referee #1 · 5 Oct 2016

In this paper the authors use terrestrial photogrammetry of a small area (30 by 30 m) to measure snow depth (h) and the associated snow cover fraction (SCF). They then use the results to define accumulation and four styles of depletion (DC) curves. These curves are ingested into a snow melt evolution model in a way that updates the fraction of area over which the model is applied. The results are found to improve the model performance.

This is a clearly written paper and a nice tidy study. It has two main conclusions which need to be explored a bit more. The first is that there were 4 styles of melt over the domain and these were a function of the antecedent history of snowfall as well as the time of year. That is an interesting and potentially useful finding, but it was derived for a near postage-stamp sized domain. So the question is "How robust and general is the finding?" Are the four styles universal for the Sierra Nevada, all

snow landscapes, or just the local area? The current set of conclusions are really a reiteration of the abstract...instead the conclusions should be about these styles and what they might mean in a more general way. That would make the conclusion relevant to snow researchers not working in the Sierra Nevada. It would also greatly strengthen the paper if the authors had any data from nearby (but not in the training area) domains that could be used to validate that the DC styles have at least local widespread validity. I have reason to think they might.

The second finding is that the ingestion of the DCs into the melt model improved the model. This is not totally novel, nor is it surprising, but it is useful. I would have liked to see a bit more quantitative assessment of the extent of the improvement. The metrics are all there, but, for example, how much better would the improvement have been if a sigle style of DC was used for all cases.

So in summary, I find this paper worth publishing, but I would ask that the authors revise the text in ways that address in greater detail the styles of melt depletion observed, whether those styles can be extended beyond the training domain, and if so, how far, and delve a little deeper into just how much improvement the ingestion of the DCs made to the model (for example, what if the model was just adjusted with a fixed linear depletion...would it compete well with the 4 styles?).

Detailed comments follow:

Page 1, Line 31: I disagree: cold northern regions also have extremely heterogeneous snow covers due to both wind-drifting and canopy interception.

Page 2, Line 5: There could be considerably more discussion in the text on the microtopography of the domain. With respect to the above comment, see: See Sturm & Holmgren (1994). Effects of microtopography on texture, temperature and heat flow in Arctic and sub-Arctic snow. Annals of Glaciology, 19(1). You will see there that northern landscapes also have heterogeneous snow.

Page 2, line 26: Here the authors speak of processes. . ...this is a good lead in to what is needed in the expanded discussion about the 4 DCs: what processes of melt (and albedo etc.) are different in each curve and why? Perhaps a table of these processes differences would be useful. They have made a good start on this in the current text, but have not really made a succinct summary of the styles and the reasons for them.

Page 5, Equation 2: It is not totally clear from the text (or the figure) how these regressions of mean depth vs. pole depth were derived. It is not even clear how many poles were in view and measured. Some additional details would help here.

Page 6, Line 11: In the equation precipitation is R not P. Correct.

Page 8, Lines 10 to 20: Good. More of this is needed.

Page 10, Line 15 and Page 12, Line 5: These assertions have no backup. . .no evidence that they are true. It would be very helpful to show that this is the case. . .or at least that there is some evidence it is true. Just stating so doesn't make it so.

Figure 5: Axis Labels are too fuzzy and small to read. Also, add a table (or schematic) that summarizes how the 4 melt curves differ and why.

Figure 6: The decision diamonds in the figure don't seem quite right. For example, what if the snow is >0.6 m and older than 30 days? This figure needs some more thought and revision.

Figure 9: Nice figure. I think looking at it, some of the physical reasons for the 4 DC styles are suggested.

---

## Referee Comment (RC2) · J. Parajka (Referee) · 20 Oct 2016

General comments

The manuscript presents parametrization of the snow depletion curves at a small plot in Mediterranean site. This parametrization is based on terrestrial time-lapse photography. The fitted curves are then implemented in a point snow model and tested for estimation of mean snow height and snow cover fraction.

Overall, the topic is interesting and within the scope of HESS. It is worth to publish, but I agree with the reviewer #1 that some revision is needed. I would also suggest to improve the discussion of the paper, relating the findings to existing literature, as well as to discuss the transferability of results to other regions. I would like to suggest to show the value of implementing this new approach into a snow model more clearly, e.g.

by comparing the simulations with a "standard" approach.

Overall I would recommend to publish the paper after some moderate revision.

Specific comments

1) Abstract: What is the GIS-based representation of snow?

2) P.2, l.20: Please correct the typo in the name Kolbert.

3) Table 5 caption does not explain the table well. The meaning (or reference) of table headers is not explained.

4) "Each cycle corresponds to the time period between the beginning of a snowfall and the end of the associated melting. Not clear what is the meaning of associated melting.". Perhaps it would be more clear if Fig.4 does indicate the start and end of selected cycles.

5) Fig.3: Would it be possible to indicate the position of rods in bottom panels?

6) Fig.5: The fit of Curve0 seems not be very close to the observations. Does this difference affect the model performance in accumulation phase? If yes, for which events and how?

7) The length of cycles is confusing "The number of cycles and their duration varied considerably over the years, with a mean number of 18 +- 5 cycles per year and a mean duration of 49+-14 and 108+- 18 days for the accumulation and melting phases of each cycle, respectively." This reads like e.g. 18 cycles per year, each has 49 days, so it is 18x49 days in a year?

---

## Referee Comment (RC3) · Anonymous Referee #3 · 24 Oct 2016

The authors analysed snow accumulation and melt cycles in four winters to determine the representative snow depletion curves which were then applied in an energy based snow accumulation and melt model. Model validation showed that the approach resulted in a reasonable reproduction of snow depth variability in an environment characterized by frequent accumulation-melt events. I am not sure that the approach which was not validated for more and larger control areas is so easily transferable to larger scales, e.g. a catchment. However, the manuscript is inspirative and I agree with its publication after modifications considering the following comments:

- Please write what is the resolution and accuracy of snow depth values estimated from the photographs and snow poles.

- Equation 2 describes the relationship beween href and h (for the whole control area).

However, two snow poles are mentioned in data description. It is not clear if data from both poles were used in calculation of h and how.

- Please correct the error in the last two lines of table 2; do they not give "mean duration", but total number of days with snow accumulation and snow melt in particular winters, respectively (this corresponds to the mean values given on p. 7, l. 16).

- Caption of Fig. 5 b and elsewhere in the text (e.g. page 8, line 5; p. 10., l.2) - the curve for snow accumulation (Curve 0) should not be called "depletion" curve, because the snow covered area is not decreasing, but increasing

- Table 3 - please add a column with number of points (days) which were available for fitting in each cycle

- Quality of Fig. 5 is not good, especially Fig. 5b is not readable

- p. 9 l. 19 - Please change formulation of the sentence - simulation of snow cover FRACTION should not have ME, MAE and RMSE in millimeters.

- Fig. 8 - adding measured precipitation and air temperature to the figures would aid explanation of the mismatches mentioned in the Discussion (p. 10., l. 24-32).

- The Discussion could also compare the results of presented approach with other works devoted to incorporation of subgrid distribution into snow models

---

## Short Comment (SC1) · 30 Oct 2016

"Note to the editor and authors: As part of an introductory course to the Master programme Earth & Environment at Wageningen University, students get the assignment to review a scientific paper. Since several years, students have been reviewing papers that are in open online discussion for HESS, and they have been asked to submit their reports to the discussion in order to help the review process. While these reports are written as official reviews, they were not requested for by the editor, and we leave it up to the editor and authors to use these reports to their advantage. While several students were asked to review the same paper, this was not done to provide the authors with much extra work. We hope that these reports will positively contribute to the scientific discussion and to the quality of papers published in HESS. This report was supervised by dr. Ryan Teuling."

Summary and recommendations

To perform hydrological modelling which is GIS-based subgrid variability is important to be taken into account. Snow coverage in Mediterranean mountainous areas is highly variable. One crucial factor in the spatiotemporal distribution of snow is micro-topography. Due to the heterogeneous snow distribution it is hard to calculate a snow depletion curve for semiarid regions which is representative for the whole area. Do define depletion curves snow cover fraction data over the whole study area is needed. Terrestrial photography was used in this research to derive snow depth and snow cover fraction at subgrid scales. In this study a four year series (the hydrological years 2009-2013) of terrestrial photography images were taken in a study area in the Sierra Nevada Mountains in Spain. The images showed a high variability in snow accumulation-melting cycles. Based on these terrestrial photography images one curve was defined for the accumulation phase of the snow cycle and four curves were derived which are representative for the melting phases. All these four cycles are representative for different kinds of melting phases. Next a sigmoid flexible function was used to parameterize the snow depletion curves. The parameterized depletion curves were implemented in the point physical snow model described in Herrero et al. (2009) so that snow cover fraction becomes a new state variable. Simulation were performed with this model to check what the optimal model parameters were. The simulation with the best parameters shows that snow depth and snow cover fraction are simulated with a high level of accuracy. Although snow cover fraction can be overestimated in short snow accumulation-melting cycles and snow depth simulation results show different results than in reality for certain states the simulation is still accurate enough. Overall it is concluded that implementation of depletion curves that are derived from terrestrial photography improves the performance of the snow model.

In my opinion the manuscript fits perfectly to the scope of Hydrology and Earth System Sciences. The research topic is very inspiring since it is a well performed research on a small scale study area. The topic is very interesting and the research continues on

previous research in the Sierra Nevada Mountains. They usage of depletion curves to improve the snow model is good since they are developed to characterize the decrease in snow cover fraction when snow melts (Luce and Tarboton, 2004; Rango and Martinec, 1982) and snow cover fraction is one of the important variables that is being studied in this research. Next to this I like the writing style of sections 4.2 and 4.3 very much. The results are discussed stepwise and in good detail. These sections are clearly written and they are well supported with figures and tables. The part of the discussion were the success of the depletion curves derived by the terrestrial photography is discussed is pleasant and clear to read. The manuscript is worth publishing after some moderate revisions. I have three main comments which I will discuss in detail in the main comment section. First of all the overall message why it is so important to fulfil this research is missing. Secondly a comparison with existing literature is missing in the discussion. The conclusion that the implementation of the depletion curves in the snow model improves it performance misses comparison with quantitative results from previous studies. Next to this I also think that the methodology of the incorporation of the depletion curves in the snow model misses some explanation.

Main comments

The overall message why it is so important to fulfill this research is missing. In general the research is a good research which is performed with a lot of high quality techniques. These techniques are all very well integrated. However the importance of the research is not well described in the introduction. Only after reading the last paragraph of the conclusion I could make a suggestion of the importance of the study. Even though it is important to note down the importance of this research in the scientific world. By doing this the readers can be more convinced about the importance of this research. I would suggest to point out in the introduction what the advantages are of the extension of point physical snow models with depletion curves. Why is it so important to know how snow cover fraction changes on such a small scale? For example Anderton et al. (2004) tells that snow models are important in hydrological modelling and

spatial variability has large influence on the prediction of surface runoff in a catchment. Another example which explains the importance of the understanding of snow cover fraction processes is that snowmelt which leads to surface runoff can only occur where the surface is covered with snow (Luce and Tarboton. 2004). I recommend to include sentences like this in the introduction and add at the end of the introduction a small paragraph where the author explains to the reader that he hopes that his results will lead to a more improved prediction of the snow cover fraction to improve the prediction of surface runoff in hydrological models.

My second main comment is about the fact that the discussion is missing comparison with previous research. The description of possible errors sources for the simulation of the model is not supported with any existing literature. It would be nice to see that the conclusions made in this paragraph are supported with previously written scientific work. Another example is that it is concluded that the results of this research are better than the results of Pimentel et al.(2015). Quantitative numbers of the accuracy of both snow model performances are not given which makes this statement hard to believe. Not only would it be better to discuss this with quantitatively results but believability of the conclusion can also increase if this research is compared to more than just one study were the performance of snow modelling is tested.

The explanation of the implementation of depletion curves in the point physical snow model is not clear enough. Right now it is impossible for other researchers to repeat this research. It is only described that the point physical is expanded by incorporating the depletion curves obtained from the terrestrial photography. However it is not explained how this is done, only that by doing this snow cover fraction will become a new state variable. Now readers can put question marks to the methodology. An unclear methodology can question the credibility of the results for the simulation of the snow model. I would suggest to explain how the depletion curves can be included in the model. Luce and Tarboton (2004) and Luce and Tarboton (1999) show that this can be done with only one sentence; by telling that the equations for the mass balance

will only be applied for the snow covered fraction. By adding a sentence like this it is clear how the implementation of the depletion curves lead to snow cover fraction as a new state variable . Next to this it was also discussed in Herrero et al. (2009) that the importance of the time step of the algorithm that solves the mass and energy fluxes is important. By including a sentence were you refer that the same algorithms as in Herrero et al. (2009) are used or an explanation why a different time step is used this problem can be solved.

Minor comments

Minor comment 1: In the text the five curves are most of the time all called depletion curves (for example p8, line 5), although this is not true for curve 0 since it represent the accumulation phase. Try to call only the four melting curves depletion curves and rather call curve 0 the accumulation curve.

Minor comment 2: The terms micro-topography or microtopography are both used in the manuscript I would suggest to use only one of them. On page 11 in line 17 it is the only time that is has been written as microtopography.

Minor comment 3: The same as above try to stay consistent with the use of accumulation/melting cycles and accumulation-melting cycles.

Minor comment 4: The reference of Luce et al. (1999) in the reference list is not published in 1999 but in 1997. This paper is used as a reference to show that in the past depletion curves have been implemented in snow physical models, but in Luce et al. (1997) depletion curves are not mentioned at all. After some research I found the paper "Sub-grid parameterization of snow distribution for an energy and mass balance snow cover model" by Luce et al (1999). This paper is published in 1999 and in this research implementation of depletion curves in snow models is discussed. I would recommend to take a look at this and include this paper in your reference list and leave out the paper by Luce et al. (1997) since it is not used anywhere else in the manuscript.

Minor comment 5: There are too many tables and figures. In the list of minor issues I have noted recommendations to combine some tables and figures. I made also a suggestion to take out one table.

List of minor issues

P1, line 30: A used reference is Luce and Tarboton (1996). In the reference list there is no reference from Luce and Tarboton from this year. So either the year number on page 1 is not correct or the reference is missing in the list.

P2, line 20: The reference Kolbert et al. (2006) should be Kolberg et al. (2006).

P3, lines 3-5: I suggest to not include this in the introduction since it is detailed methodology. The first introduction of the rods should be mentioned in the study site and available data section.

P4, line 22: There is a typo in the reference Ying et al. (2003). It should be Yin et al (2003).

P5, lines 6-7: This should not be included in the methodology section. The fact that 5 parameterizations are selected is a result. Therefore I suggest to remove these lines from this section.

P5, line 19: The previous defined linear equation is not defined earlier in the text. This arises question of the credibility of your methodology section. Provide in section 3.2 also a description of this equation that is used.

P6, lines 8 and 11: In equation 3 precipitation is defined with R, however in the text it is described that P defines the precipitation.

P6, line 16: Include a reference that support why W can be disregarded. In Pimentel et al. (2016) it is stated that the rapid snow metamorphosis is observed but also in this paper a proper reference is missing. I would recommend to refer to the observations or give another proper reference.
P7, line 2: The year of the reference Cline (1999) mismatches the year in the reference list.

P7, lines 15-16: It is stated that each year has a mean number of 18± 5 cycles a year and a mean duration of 49±14 days for the accumulation phase and 108 ±18 days for the melting phase. This tells me that one cycle has a mean duration of 49+108 = 157 days. But 18 times 157 days will exceed the amount of days in one year. This is very confusing.

Table 1: This table can be left out. I did not have the idea that this gives important information needed for the research. Rather describe more of the meteorological conditions in the text.

Table 3 and 4: Add table together since they both give information about the fitted parameters for the depletion curves.

Table 5 and 6: Figure 6 is a very small table which continues on Table 5 since it shows the errors of the calibration parameters used in simulation 7. I would recommend to add Table 6 with Table 5.

Figure 3: This figure is used to show that terrestrial photography can be used to determine accurate depletion curves even when the atmospheric conditions are different. Include in the caption which atmospheric conditions were present for which picture. This makes the argument more powerful.

Figure 5: The quality of this figure is not good. The upper part with the accumulation-melting cycles is still readable, however a higher resolution would give a more neat figure. The quality of the depletion curve patterns in part b is very bad. Since I can see the shape of the accumulation curve and the depletion curves the fact that the 5 curves are very different was still clear to me. But I am not able to read the axes and titles. Due to this detailed discussion of the results cannot be followed.

Figure 4 and Figure 5: Since part a of Figure 5 and Figure 4 both show the

accumulation-melting cycles for the calibration periods I would suggest to combine these two figures. In Figure 4 the lines of the snow depth for the cycles that are used for the calibration process can be made bold. I would recommend to add the depletion curves of part b of

Figure 5 underneath the calibration and validation figures.

Figure 6: The same accounts as for Figure 5; the depletion curves figures have a low resolution, which makes it impossible to read the figures properly.

References

Anderton, S.P., White, S.M. and Alvera, B.: Evaluation of spatial variability in snow water equivalent for a high mountain catchment, Hydrological processes, 18, 435-453, 2004

Herrero, J., Polo, M.J., Moñino, A. and Losada, M.A.: An energy balance snowmelt model in a Mediterranean site, J.Hydrol., 371, 98-107, 2009

Luce, C.H., Tarboton, D.G. and Cooley, K.R.: Spatially distributed snowmelt inputs to a semi-arid mountain watershed, Proceeding of the Western Snow Conference, Banff, Canada, 1997

Luce, C.H., Tarboton, D.G. and Cooley, K.R: Sub-grid parameterization of snow distribution for an energy and mass balance snow cover model, hydrological processes, 13, 1921-1933, 1999

Luce, C.H., Tarboton, D.G.: The application of depletion curves for parameterization of subgrid variability of snow, Hydrological processes, 18, 1409-1422, 2004

Pimentel, R., Aguilar, C., Herrero, J., Pérez-Palazón, M.J. and Polo, M.J.: Comparison between Snow Albedo Obtained from Landsat TM, ETM+ imagery and the SPOT VEGETATION Albedo Product in a Mediterranean Mountainous Site, Hydrology, 2016

Rango, A. and Martinec, J.: Snow accumulation derived from modified depletion curves

of snow coverage, International Association of Hydological sciences, Proceedings of the Exeter Symposium, 1982, Publ.no 138

---

## Short Comment (SC2) · 31 Oct 2016

"Note to the editor and authors: As part of an introductory course to the Master pro-gramme Earth & Environment at Wageningen University, students get the assignment to review a scientific paper. Since several years, students have been reviewing papers that are in open online discussion for HESS, and they have been asked to submit their reports to the discussion in order to help the review process. While these reports are written as official reviews, they were not requested for by the editor, and we leave it up to the editor and authors to use these reports to their advantage. While several students were asked to review the same paper, this was not done to provide the au-thors with much extra work. We hope that these reports will positively contribute to the scientific discussion and to the quality of papers published in HESS. This report was supervised by dr. Ryan Teuling."

Summary

Using the combination of terrestrial photography (TP) and snow depletion curves this paper aims for a new way to describe subgrid variability of snow distribution in the Sierra Nevada. Because of the changeable climate conditions in this area, multiple ac-cumulation and melting cycles occur throughout the snow season. Using data for snow cover fraction (SCF) and snow depth (h) from a previous four-year study (Pimentel et al., 2015), five different depletion curves (one for accumulation, four for melting) were parametrized using a flexible sigmoid function adopted from Yin et al. (2003). Subse-quently, these depletion curves were implemented into a point snow model developed by Herrero et al. (2009) with the use of a decision tree. Using three years for calibra-tion and one year for validation, simulations of the SCF and snow depth were found to be generally accurate compared to the observed values. Although the simulations in both calibration and validation periods showed an overestimation of the SCF and a mismatch in snow depth values in some states, the use of TP allowed for errors in the simulations to be related to potential error sources in measurements.

Recommendations

This paper is well written and structured which allows for an easy understanding of the methods used and results obtained. The introduction gives a clear overview of previous research in this field and the authors do a good job in explaining the novelty of the approach. I think the paper fits the scope of HESS particularly well because of the multidisciplinary approach, which is shown by combining the products of previ-ous studies in order to derive new knowledge and conclusions. The use of terrestrial photography shows that cost effective methods can be integrated with a modelling ap-proach. Although I feel this paper is almost ready for publication I think the discussion in particular needs more elaboration in order to illustrate that the methods used and results obtained are an addition to the field of snow distribution modelling.

General Comments

The main problem I have with this paper is that, although the approach is novel, this is not emphasized enough in the discussion. The introduction outlines flaws in the methods of previous attempts to capture subgrid variability and argues the paper's novel approach could be an improvement. The performance of the produced model was compared to the field observations to check its' general accuracy, but this is however not enough back up the claim of novelty in the introduction. The whole study is based on the expansion of the model which was developed by Herrero et al. (2009), but no comparison of the supposedly improved model is offered with the original model. There is a mention that the inclusion of the 5-curve set of DC's improved the model (page 10, line 17), this however remains quite vague while I think this should be one of the main results of the study. What would help is including a section where the new model is directly compared with the previous model by simulating the same period with both models. This quantification also allows the reader to decide for themselves if the increased accuracy is justified by the increase in effort in respect to future research.

Additionally, I think there should also be a comparison with different sets of DC's. The text mentions that the DC's are clustered if parameter values are within 10% of each other (page 5, line 4). While there is merit in clustering the DC's like this, no argumentation is given for this value. A larger value would obviously lead to less depletion curves and different simulation results. How would including only one depletion curve for accumulation and one for melting affect the model results? Increasing the number of parameters and complexity of a model will almost always improve the results but to what extend will this affect the application of this method in other areas. I think it would be useful to include a section on this in the discussion.

The introduction also mentions the problem of defining a DC for a whole watershed because of the large spatial and temporal variability in snow distribution (page 2, line 23). The authors state that using a distributed application of DC's could be used to capture this variability. However, the control area used in this study was relatively small (30mx30m) and while the conclusion proposes that this method provides a foundation

for the extension of snow point models, the discussion didn't contain information about the usefulness of this method on larger scale. Could this method be applied in large areas such as a whole watershed? For example, Luce et al. (2004) defined depletion curves for the Upper Sheep Creek basin which is approximately 26 ha in size. They did this for one accumulation-melting cycle which is more common for higher latitudes, as was mentioned in the introduction. In contrast to your findings, they found little variability between years. Would this also be the case for areas such as the sierra Nevada, where multiple accumulation-melting cycles occur throughout the year. I also wonder how the terrestrial photography method could be applied in larger areas. Was the high resolution essential for the results? Would remote sensing also have worked in this situation as was done in Kolberg et al. (2006)? I think it would be useful if these upscaling issues were mentioned in the discussion. In this way, the discussion would be more in line with what the introduction stated the approach and goals of this research would be.

Specific Comments

I think the title should mention the use of snow depletion curves as it is a major part of this study.

The RMSE of the simulations is mentioned in the abstract but these are the RMSE values for the calibration period. I think showing the RMSE values that were found during the validation of the model would be more representative of the general accuracy.

Page 2, line 12: "(Mark and Dozier, 1992)" should be "(Marks and Dozier, 1992)".

Page 2, line 20: "Kolbert et al. (2006)" should be "Kolberg et al. (2006)".

Page 4, line 15: The explanation of what exactly constitutes an accumulation melting cycle remains vague to me. Does a cycle mean the time between the accumulation of snow from a certain level and when it returns to that same level due to melting? Is it possible for multiple cycles to be ongoing at the same time? Please elaborate.

Page 4, line 22: "Ying et al. (2003)" should be "Yin. et al. (2003)".

Page 5, line 4: How did you decide upon using 10% difference for clustering the curves? There should be an argumentation for this.

Page 5, line 17: The explanation of how snow depth was obtained from the photos with the clustering algorithm needs more elaboration. Was it necessary to use two snow rods if the reference snow depth was used to calculate the actual average snow depth? What was the previously defined linear equation mentioned in line 19?

Figure 2 implies the relation between the average snow depth and the reference snow depth is linear, which it is not. The figure could be clearer on explaining the relation between the reference and average snow depth.

Page 6, line 11: In equation 3, R is used to indicate the precipitation while the text says P.

Page 7, line 21: Did excluding the cycles with short duration and cycles were the area was completely covered leave you with 16 cycles? Could use a better explanation on the method of choosing the cycles that were eventually used for calibration. It does seem from figure 5a that cycles were only used for calibration if the SCF dropped to 0. This was however not mentioned in the text.

Table 2: In the 6th and 7th row of the table: instead of showing the mean duration of accumulation and melting cycles for the whole study period, the means of all columns are summed as was done for the 5th row.

Figure 5b: The axes on the graphs of the curves are not clear.

Table 4: The values of h(e) in the table don't correspond with the values in the text at page 8, line 21.

Table 3: I think it would be practical if the table was sorted according to the curve types.

References

Herrero, J., Polo, M.J., Moñino, A. and Losada, M.A.: An energy balance snowmelt model in a Mediterranean site, J.Hydrol., 371, 98-107, 2009

Kolberg, S. A., and Gottschalk, L.: Updating of snow depletion curve with remote sensing data, Hydrol. Process., 20, 2363-2380, 2006.

Luce, C.H., Tarboton, D.G.: The application of depletion curves for parameterization of subgrid variability of snow, Hydrological processes, 18, 1409-1422, 2004

---

## Short Comment (SC3) · 2 Nov 2016

"Note to the editor and authors: As part of an introductory course to the Master programme Earth & Environment at Wageningen University, students get the assignment to review a scientific paper. Since several years, students have been reviewing papers that are in open online discussion for HESS, and they have been asked to submit their reports to the discussion in order to help the review process. While these reports are written as official reviews, they were not requested for by the editor, and we leave it up to the editor and authors to use these reports to their advantage. While several students were asked to review the same paper, this was not done to provide the authors with much extra work. We hope that these reports will positively contribute to the scientific discussion and to the quality of papers published in HESS. This report was supervised by dr. Ryan Teuling."

This paper has derived several depletion curves and implemented them to upscale a point model to a larger grid. To do this the snow cover fraction and the snow depth are obtained using terrestrial photography using the methods from Pimentel et al.(2015). From that data and with a sigmoid function, 5 depletion curves were derived, one curve for accumulation and four curves for the melting of snow. These depletion curves were then implemented into a point snow model from Herrero et al.(2009), through a decision tree. The model was calibrated with different simulations over 3 years and after that a validation run was done for one year with the optimal simulation. The results show an accurate SCF and snow depth with similar results for both the calibration and validation. There are some errors due to weather phenomena that are not implemented into the model. The use of different depletion curves makes upscaling to larger areas possible.

I think this paper is appropriate for the journal of hydrology and earth system science because the modeling of snow is an important part of the hydrological modeling and it fits well with the scope of the paper. The research is new and innovating. The use of several depletion curves to describe accumulation and melting has never been done before and instead of the WUE like most papers, the snow depth was used for the depletion curves. This derivation of these depletion curves from terrestrial photography is an innovative approach that will be useful in further research.

This paper provides a very good research with solid methods. The methods chosen fit well together and form a consistent research together. They are well implemented from the previous literature and still very well written so that it is understandable how the methods are applied. They are to the point and explained well. Also this paper gives a very structured and good presentation of results. It is really understandable what they have done in the methods and how they came to their results. The results are given for every step of the methods, throughout the process of deriving the curves and implementing the model. This makes the process understandable and repeatable and the results credible. By chopping up the results in reasonable parts, it has a good structure which makes it nice to read. The figures and tables of the results also are understandable and are easy to understand. The introduction gives a good indication of previous methods used and it gives a good overview of history of snow modeling, making the subject more understandable The writing style of the paper is one of the lesser parts of this paper. It has been written like all the background information is already known to the reader and more details need to be implemented for better understanding. Also the discussion could be improved in several ways. Overall, I think that the research done in this paper is really well thought out and very well executed. Therefore I recommend publication for this paper with some moderate revisions following the arguments below.

First I will explain the major arguments that I think must be addressed, afterwards there are my minor comments of how to improve the paper and lastly there is a list of minor revisions.

The introduction starts with the main reason for this research. Thereafter it goes into detail of what is known and unknown and the actual goal of the research. After reading the introduction it was difficult to understand the paper since there is no context. What is missing here is an introduction of the subject of the paper, which is important for people who are no expert in the field of snow modeling. In this paper, there is only one sentence introduction(p1, line 25) and no context given. The reason for initiating this research is too short for people who have not read the background material. It is unclear from the introduction why the scale issues are still an issue since Bloschl et al is written in 1999 and why this snow distribution modeling is important at all. This change will only affect the introduction of the paper, which can be improved in some ways. One option is a single figure in the introduction that gives the context of the subject by introducing the links between the most used terms. An explanation of the subject is also possible. In Anderton et al,(2004) and Luce et Tarboton,(2004) they first introduce the subject of snow models and explain various terms. Then they take a narrower view towards their problem that they want to solve. To broaden the reason for this research, more detail could be applied. Bloschl et al, 1999 came first with the

problem of scale issues, but in this introduction there is not yet an explanation what these scale issues pertain or why they are important in the context of snow modeling so that can be implemented in the introduction.

In the discussion, the results are only compared to one other study that is a previous study using the same method, namely Pimentel et al.(2015). This is also only done in a qualitative setting by mentioning that the results are improved without mentioning what the results were of the previous study(P10, line 17-19). By not comparing the results with other papers using different methods of modeling subgrid variability, the credibility of the results can be questioned. It undermines the feasibility of this research since no quantitative comparisons are done. The addition of numbers will give a better visual presentation and makes comparison much easier for the reader. Also other papers can be mentioned that used other models or methods to derive the depletion curves such as Kolberg et al(2006), Luce et Tarboton, 2004 and also Herrero et al, 2009 (other DC's). This shows how these results compare within the field of snow distribution modeling and if the model is truly a good foundation for further upscaling of models as stated in the conclusion.

One of the goals of this paper is to give an insight in the upscaling from a point model to a 30x30 grid(P6, line 25-26). However, nowhere in the paper it is explained how this upscaling is done in the model. There is a mention that this upscaling is done by implementing the depletion curves into the model (P6, line 25-26), but by indicating that, still no explanation is given for the process of the actual upscaling. By not describing this process, an important part of the methods is left out. The research is not repeatable and less applicable for further research on upscaling. This can be remedied by explaining the process of the upscaling in the methods. This can be done for example by a step by step explanation or a short summary including the formulas used as is done in Luce et al, 1999 and Pimentel et al, 2015.

The minor comments:

[Figure]

Minor comment 1) The results of the RMSE mentioned in the discussion, conclusion and abstract are the results that were achieved with the calibration of the model. This seems counterintuitive since the validation of the model indicates how well the model actually works, while the calibration gives the optimal values for the model that can be achieved. Therefore the results of the validation period are more important to indicate how well the model works and should be used instead of the calibration results. The argument is given that these results are almost the same, but then please explain on what that is based, because there is still a difference in the number and if the numbers are indeed significantly the same, then still the validation results can be used for the comparison to other papers.

Minor comment 2) The second paragraph of the discussion (line 8-11) is unclear in what is meant. Figure 9 is a important figure but does not seem understandable now. An explanation is needed of what is exactly visible in the different pictures, and what is different between the pictures with same SCF. In line 8-11 an explanation of weather phenomena are given that are not clearly visible in figure 9. Please rewrite this paragraph and give an explanation of the differences visible, and in weather conditions, in the pictures in figure 9.

Minor comment 3) The third paragraph of the discussion (line 12-16) is unclear in their goal. The arguments given in this paragraph do not seem to relate to the conclusion in the last sentence of this paragraph. Please explain this relation better and why that conclusion can be drawn. Also that conclusion relates to the last sentence in this paper, which is a conclusion that this research provides a basis for extension of snow models to larger areas. However these conclusions do not say the same thing. Please relate the two conclusions better with each other and give argument for these conclusions. Also explain why these conclusions can be drawn from the results because that is now not completely clear in the paper.

Minor comment 4) This paper has a good explanation of DCs. The description of the method of acquiring these depletion curves is very accurately described and it is well

thought out how they would use the data to derive the depletion curves. Especially page 8 gives a very clear overview of each depletion curve that is used. This part is very nice and well written.

Minor comment 5) In paragraph 3.2 the methods for acquiring the snow depth with the rods is explained. However is unclear to me how the reference snow depth is calculated. Figure 2 does not help in explaining this method. Please give more explanation of the method and explain figure 2 in more detail. Also why the different snow depths are visible at the levels that they are located on.

Minor comment 6) At the very end of section 4.2 on page 8 there is said that a decision tree is implemented into the model, but it is not described how this is done. The decision tree is not even mentioned in the methods. Please describe in the methods how this decision tree is implemented in the model.

Minor comment 7) At the end of page 10, the error sources of the depletion curves are given. Both the first and third error also give a possible explanation for these errors. However the second error does not have any explanation of how this insufficiency of rain-over-snow effects is caused. Please indicate the reason for this error, or indicate that it is unclear how this error is caused.

Minor revisions:

P1, line 17-18: "The resulting DCs were able to capture certain physical features of the snow, which were used..." seems like the physical features were used and included into the model instead of the depletion curves, so the sentence structure could be changed.

P1, line 29-30:" Luce et Tarboton, 1996" is not mentioned as such in the references, should be Tarboton et Luce, 1996 or reference needs to be changed.

P2, line 20: "Korbert" should be "Kolberg".

P2, line 28-31: These sentences can be moved to the methods, since they describe

the possible shapes for depletion curves.

P3, line 3-5: These sentences can be moved to the methods, since this information is missing there, and if it all information on the methods of the rods is put together, the explanation of the rods is more understandable.

P4, line 22:"Ying et al" should be "Yin et al".

P5, line 6-7: This sentence pertains results and does not have to be mentioned in the methods.

P5, line 19: "A previous defined function" has not been defined in this paper, so please put it in the paper.

P6, line 11: "P" is not mentioned in formula, should be "R" that is mentioned in formula above, or the "R" should be changed into a "P".

P7, line 15-16: The paper mentions 18 cycles per year with a duration of 49+108=157 days for each cycle. This seems very illogical since it seems that the cycle duration is too long to fit 18 times in one year. Either a bit more explanation that cycles can overlap is necessary or the sentence structure needs to be changed.

P7, line 26-28: These sentences can be moved to the methods. The amount of detail given here belongs in the methods, not in the results.

P8, line 28-29: This sentence belongs in the methods, since it is not mentioned there.

P9, line 2-4: These sentences belong in the methods. The decision tree is not mentioned in the methods and this amount of detail should be mentioned in the methods.

P11, line 3: it says "error sources of error" so one error can be left out.

P11, line 28: It says "error of less than", but in the rest of the paper these amounts are given as the error, not a smaller value as is insinuated here.

P14, line 5-6: I think the wrong reference title is mentioned here, since the paper with

that title is from 1997, and does not mention depletion curves, while the paper that is referred too does mention DCs as indicated in the text P2 line 14.

P16, table 1: This table can be left out of the paper, since this figure does not have added value to the understandability of the paper.

P17, table 2: This table can be left out of the paper, since this information is not necessary to understand the paper, and is not explained in the text.

P 20/21, table 5 & 6: these figures can be combined into one figure since table 6 is very small.

P 23, fig 2: it is unclear how href and h1 and h2 came to be in this figure, more explanation in the header can give more understanding of what is meant with each parameter visible in the figure.

P24, fig 3: The header of the figure could use an explanation about the differences in weather condition between the three different dates, as it is now quite unclear what makes them different since now especially the first and last date seem to indicate the same conditions.

P 25/26 fig 4 & 5: both figures can be combined, the only difference in figure 4 with 5 is that all cycles are mentioned in figure 4 instead of only the cycles used in this paper as in figure 5. Placing two of the same figures in the paper with little difference in information is unnecessary.

P26, fig 5: axis of the DC curves are not readable.

P27, fig 6: axis of the DC curves are not readable.

P27, fig 6 "More than 30 days with previous snow" is unclear what is meant, so it would be good to reformulate.

---

## Author Comment (AC1) · 13 Dec 2016

The comment was uploaded in the form of a supplement: http://www.hydrol-earth-syst-sci-discuss.net/hess-2016-426/hess-2016-426-AC1-supplement.zip

---

## Author Comment (AC2) · 13 Dec 2016

The comment was uploaded in the form of a supplement:
http://www.hydrol-earth-syst-sci-discuss.net/hess-2016-426/hess-2016-426-AC2-supplement.zip

---

## Author Comment (AC3) · 13 Dec 2016

The comment was uploaded in the form of a supplement:
http://www.hydrol-earth-syst-sci-discuss.net/hess-2016-426/hess-2016-426-AC3-
supplement.zip

---

## Author Comment (AC4) · 13 Dec 2016

The comment was uploaded in the form of a supplement:
http://www.hydrol-earth-syst-sci-discuss.net/hess-2016-426/hess-2016-426-AC4-
supplement.zip

---

## Author Comment (AC5) · 13 Dec 2016

The comment was uploaded in the form of a supplement:
http://www.hydrol-earth-syst-sci-discuss.net/hess-2016-426/hess-2016-426-AC5-supplement.zip

---

## Author Comment (AC6) · 13 Dec 2016

The comment was uploaded in the form of a supplement:
http://www.hydrol-earth-syst-sci-discuss.net/hess-2016-426/hess-2016-426-AC6-supplement.zip

---

## Author Comment (AC7) · 13 Dec 2016

The comment was uploaded in the form of a supplement:
http://www.hydrol-earth-syst-sci-discuss.net/hess-2016-426/hess-2016-426-AC7-supplement.zip

---

## Author Comment (AC8) · 13 Dec 2016

The comment was uploaded in the form of a supplement:
http://www.hydrol-earth-syst-sci-discuss.net/hess-2016-426/hess-2016-426-AC8-supplement.zip

---

## Author Response (AR1)

**COMMENTS TO REFEREE #1**

**In this paper the authors use terrestrial photogrammetry of a small area (30 by 30 m) to measure snow depth (h) and the associated snow cover fraction (SCF). They then use the results to define accumulation and four styles of depletion (DC) curves. These curves are ingested into a snow melt evolution model in a way that updates the fraction of area over which the model is applied. The results are found to improve the model performance.**

**This is a clearly written paper and a nice tidy study. It has two main conclusions which need to be explored a bit more.**

**The first is that there were 4 styles of melt over the domain and these were a function of the antecedent history of snowfall as well as the time of year. That is an interesting and potentially useful finding, but it was derived for a near postage-stamp sized domain. So the question is "How robust and general is the finding?" Are the four styles universal for the Sierra Nevada, all snow landscapes, or just the local area? The current set of conclusions is really a reiteration of the abstract… instead the conclusions should be about these styles and what they might mean in a more general way. That would make the conclusion relevant to snow researchers not working in the Sierra Nevada. It would also greatly strengthen the paper if the authors had any data from nearby (but not in the training area) domains that could be used to validate that the DC styles have at least local widespread validity. I have reason to think they might.**

**The second finding is that the ingestion of the DCs into the melt model improved the model. This is not totally novel, nor is it surprising, but it is useful. I would have liked to see a bit more quantitative assessment of the extent of the improvement. The metrics are all there, but, for example, how much better would the improvement have been if a single style of DC was used for all cases. So in summary, I find this paper worth publishing, but I would ask that the authors revise the text in ways that address in greater detail the styles of melt depletion observed, whether those styles can be extended beyond the training domain, and if so, how far, and delve a little deeper into just how much improvement the ingestion of the DCs made to the model (for example, what if the model was just adjusted with a fixed linear depletion...would it compete well with the 4 styles?).**

First of all, we would like to thank Referee #1 his comments and suggestions on the paper. The authors fully agree with Referee #1 in the validation of the findings at different locations over Sierra Nevada. In fact, replications of the monitoring system, that is, camera+weather station, have been recently installed in selected sites throughout the area of Sierra Nevada, but there are not data series long enough so far to perform the analysis. Other data sources such as remote sensing and field measurements of snow depth values were used in a previous effort to this work with the terrestrial images, but the scarcity of available measurements and the cloudy conditions during most of the periods of interest (with snow presence) made us focus of ground sources. Nevertheless, we think that the results of this work lead to relevant conclusions under the scope of its goals: to incorporate the effects of the spatial variability of the snow distribution at subgrid scale for a Mediterranean site. The subgrid scale effects may affect significantly the extension to a raster-model of the physical modeling of the snow, especially in regions where both the spatial and time variability are high, in two ways: first, when applying the physical equations from point to cell-size calculations; this is what we are focusing on in this work. Secondly, the application of these results in larger areas; this is what we are working on at the moment.

With this in mind, despite the DCs were obtained over a 30x30m area as control point, there are some results that allow us to conclude on the usefulness of this approach and its potential representativeness beyond the local scale. On one hand, the four years of data in our study (3 calibration + 1 validation) exhibited a high variability of both the states in the monitoring point during each year, and the annual regime of snow, and thus they cover a wide domain of daily states of the snow distribution. The curves were described from dimensionless expressions of the selected variables, snow depth and snow cover fraction, so that they could be analyzed under the light of the physical conditions prevailing during each analyzed melting cycle and some relationship could be explored (as it turned out). The fact that we could finally explain the observed variability from four different patterns of the snowmelt dynamics associated to specific conditions, which was validated during an additional year, is an indirect but useful support to the potential application of the DCs not only in different points in Sierra Nevada, but also in other regions and/or similar snow states cycles, once the local order of magnitude of snow depth is estimated.

We tried to show this in the text and in the results showed in Figure 9. Following this comment, further explanation in the discussion has been included (see page 10, lines 19-34, and page 11, lines 1-14 in the revised version) together with a new table (Table 6) that gathers the factors and processes dominant in each pattern of melting dynamics. We have also emphasized the scope of the work and the on-going work to directly validate the general applicability of the results (see page 11, lines 25-34, in the revised version).

On the other hand, regarding the need for a single or multiple curve approach, the conclusions of a previous study (Pimentel et al., 2015) suggested that the use of a unique depletion curve was not enough to capture the complex dynamics of snow on this scale in these regions. In this study, we tested the use of a single depletion curve during the study period from the same data set. Three different parameterizations of a lognormal distribution, described by their coefficient of variation (CV=0.4, CV=0.8, CV=1.2) were assessed following the curves proposed by Luce and Tarboton (2004). The results pointed out the likely improvement to be obtained if different parameterizations were used during the snow season or depending on the year, but the use of predetermined curves did not lead to their respective association to certain conditions during the melting cycles. Our current work is the consequence, and the results show an improved performance of the snow model, with RMSE values of 84.2 and 105.8 mm during the calibration and validation periods, respectively, below the RMSE values of 321.3 mm, 285.4 mm and 556.0 mm obtained in the previous work for each CV-parameterized curve tested in the work, respectively. The performance of the multiple-DC choice in the snow model is also better than the results obtained with a single-DC choice plus the assimilation of the observed SCF values (also in Pimentel et al., 2015).

| | Single DC | | | 4 DCs | |
|---|---|---|---|---|---|
| | CV=0.4 | CV=0.8 | CV=1.2 | Calibration | Validation |
| Without assimilation | 321.26 | 285.37 | 555.96 | 84.2 | 105.8 |
| With assimilation | 279.23 | 191.18 | 523.14 | | |

Following the Referee's comment, further explanation has been included both in the Introduction and Discussion sections (see page 2, line 29, and page 12 lines 1-14, in the revised version)

**Detailed comments follow:**

**Page 1, Line 31: I disagree: cold northern regions also have extremely heterogeneous snow covers due to both wind-drifting and canopy interception.**

The authors fully agree, and the sentence has been rewritten following this comment (see page 2, lines 1-2, in the revised version).

**Page 2, Line 5: There could be considerably more discussion in the text on the microtopography of the domain. With respect to the above comment, see: See Sturm & Holmgren (1994). Effects of microtopography on texture, temperature and heat flow in Arctic and sub-Arctic snow. Annals of Glaciology, 19(1). You will see there that northern landscapes also have heterogeneous snow.**

We have included some text in this section (see page 1, lines 7-8, in the revised version)

**Page 2, line 26: Here the authors speak of processes. . ...this is a good lead in to what is needed in the expanded discussion about the 4 DCs: what processes of melt (and albedo etc.) are different in each curve and why? Perhaps a table of these processes differences would be useful. They have made a good start on this in the current text, but have not really made a succinct summary of the styles and the reasons for them.**

This is certainly a key point and we have failed to fully highlight it. Following the suggestion of the Referee #1, a new table that sums up selected descriptors of the main processes involved in each DC pattern identified (Table 6 in the revised version) has been included.

**Page 5, Equation 2: It is not totally clear from the text (or the figure) how these regressions of mean depth vs. pole depth were derived. It is not even clear how many poles were in view and measured. Some additional details would help here.**

To clarify this, we have included some text in this section (see page 5, lines 22-34, in the revised version) regarding how many rods were used, the conversion of pixel to meters, and how to derive a distributed value from the point measurements. A new version of Figure 2 associated to this has been also included.

**Page 6, Line 11: In the equation precipitation is R not P. Correct.**

This typo error has been corrected in the revised manuscript (see page 6, lines 15, in the revised version).

**Page 8, Lines 10 to 20: Good. More of this is needed.**

**Page 10, Line 15 and Page 12, Line 5: These assertions have no backup. . .no evidence that they are true. It would be very helpful to show that this is the case... or at least that there is some evidence it is true. Just stating so doesn't make it so.**

As we have commented in the initial general comment, we cannot validate directly with measurements these assertions. However, the way in which the curves were described, see previous answer to the general comments. Some clarifications have been added to the Discussion section (see page 11, lines 25-34, in the revised version).

**Figure 5: Axis Labels are too fuzzy and small to read. Also, add a table (or schematic) that summarizes how the 4 melt curves differ and why.**

The font size has been increased and a summary table has been added (see Table 6 in page 24 in the revised version).

**Figure 6: The decision diamonds in the figure don't seem quite right. For example, what if the snow is >0.6 m and older than 30 days? This figure needs some more thought and revision.**

This figure tries to explain the decision tree included in the snow model to choose a given curve during the calculation process. Each curve represents a different melting behavior which is in practice highly conditioned by the previous states of the snowpack. The condition formulated in the diamonds translate these previous snow state conditions into test variables the model is capable to check out. For example, the model does not simulate/monitor the rate of compaction, but since this process is closely related with the age of the snow this condition has been simplified by using the number of days with continuous presence of snow. The same takes place with the effect of the micro-topography, for which a reference depth above which its impacts on the snow change is negligible, since the micro-topography is completely covered by snow has been used. The numerical values for each test variable were estimated from the observations. 30 days is the maximum number of days that make the different between the Curve 1 and 2, and 0.6 m is the 95$^{th}$ percentile of the distribution of height in the micro-topography in the study area. In future extension of the model to different areas, these values are prone to change and should be adapted to particular conditions. This arrangement in the tree comes from the observations and therefore, from the importance of the process on the melting parameterized in each of them. Some clarifications have been added to the Discussion section (see page 9, lines 12-16, in the revised version).

Answering the direct question if the snow is >0.6 m and older than 30 days, the age of the snow simply prevails and the selected curve may be the 1.

**Figure 9: Nice figure. I think looking at it, some of the physical reasons for the 4 DC styles are suggested**

**COMMENTS TO REFEREE #2**

**General comments**

**The manuscript presents parametrization of the snow depletion curves at a small plot in Mediterranean site. This parametrization is based on terrestrial time-lapse photography. The fitted curves are then implemented in a point snow model and tested for estimation of mean snow height and snow cover fraction.**

**Overall, the topic is interesting and within the scope of HESS. It is worth to publish, but I agree with the reviewer #1 that some revision is needed. I would also suggest to improve the discussion of the paper, relating the findings to existing literature, as well as to discuss the transferability of results to other regions. I would like to suggest to show the value of implementing this new approach into a snow model more clearly, e.g. by comparing the simulations with a "standard" approach.**

**Overall I would recommend to publish the paper after some moderate revision.**

We would like to thank Referee 2 for the remarks and suggestions included in the revision. Following these general comments, we have included further discussion regarding the association between the identified accumulation/depletion curves and the main physical conditions during the cycles they describe. This is an indirect way for a first validation of the applicability of the curves beyond the local scale (see also the response to Referee 1). Additionally, we have included references to the context of other works regarding our results and conclusions in the Discussion section (see page 11 line 34 and page 12, lines 13, in the revised version).

Moreover, we have included in the discussion the comparison of the current results with those from a previous work in which standard depletion curves were used to test the performance of the snow model in these heterogeneous areas (see page 12, lines 1-14, in the revised version). This point was also addressed by Referee 1, and we have included further justification of the multiple-choice of curves during the same snow season to improve the model's capability to reproduce the snow dynamics.

Balk, B., Elder, K.: Combining binary decision tree and geostatistical methods to estimate snow distribution in a mountain watershed, Water Resour. Res., 36, 13–26, 2000.

Erxleben, J., Elder, K., Davis, R.: Comparison of spatial interpolation methods for estimating snow distribution in the Colorado Rocky Mountains, Hydrol. Processes, 16, 3627–3649, 2002.

Molotch, N. P., Colee, M. T., Bales, R. C., Dozier, J.: Estimating the spatial distribution of snow water equivalent in an alpine basin using binary regression tree models: The impact of digital elevation data and independent variable selection, Hydrol. Processes, 19, 1459–1479, 2005.

Molotch, N. P, Margulis, S. A.: Estimating the distribution of snow water equivalent using remotely sensed snow cover data and a spatially distributed snowmelt model: A multi-resolution, multi-sensor comparison, Ad. Water Res, 31 (11), 1503-1514, 2008.

**Specific comments**

**1) Abstract: What is the GIS-based representation of snow?**

We aimed at describing the grid-based spatial domain of distributed models to simulate the evolution of the snowpack. But we agree that this term may be unfortunate, and we have changed "GIS-based" into "grid-based" in the revised text.

**2) P.2, l.20: Please correct the typo in the name Kolbert.**

We apologize for this typo, which has been corrected in the revised version of the manuscript.

**3) Table 5 caption does not explain the table well. The meaning (or reference) of table headers is not explained.**

We have rewritten this caption (see page 22, lines 1-3, in the revised version) to clarify the information of the Table.

**4) "Each cycle corresponds to the time period between the beginning of a snowfall and the end of the associated melting. Not clear what is the meaning of associated melting.". Perhaps it would be more clear if Fig.4 does indicate the start and end of selected cycles.**

Following this, we have changed this into "…to the time period between the beginning of a snowfall event and the end of the complete ablation of the snow or the occurrence of a new snowfall event", (see page 4, lines 21-22, in the revised version).

**5) Fig.3: Would it be possible to indicate the position of rods in bottom panels?**

This information has been included in the revised version of Figure 3.

**6) Fig.5: The fit of Curve0 seems not be very close to the observations. Does this difference affect the model performance in accumulation phase? If yes, for which events and how?**

The representation of the Curve 0 in Fig.5 can be misleading since most of the dots in the figure are located very close to the line SCF*=1 and specifically near the h*=1, and then they cannot be appreciated as much as those out of this domain in the Figure. Table 4 shows the determination coefficient for this fitted curve, 0.85; this also influenced the adoption of a single curve for the accumulation phase of each cycle. However, the curve succeeded in making the model simulate well the accumulation phases during the study period, as can be observed in both Figures 6 and 7 (please, note that numbers refer to those in the revised version), with divergences occurring mostly in the reproduction of snow depth values after those melting events that were not fully captured by the model, that is, when the memory of the model biases the simulation. This can be also explained from the general pattern of the snowfall events in these Mediterranean regions, where heavy but quick snowfalls are very usual, with less variability than that exhibited by the melting phases.

Following this comment, further detail has been included in the Discussion section (see page 10, lines 23-28, in the revised version).

**7) The length of cycles is confusing "The number of cycles and their duration varied considerably over the years, with a mean number of 18 +- 5 cycles per year and a mean duration of 49+-14 and 108+- 18 days for the accumulation and melting phases of each cycle, respectively." This reads like e.g. 18 cycles per year, each has 49 days, so it is 18x49 days in a year?**

We have rewritten the sentence: The number of cycles and their duration varied considerably over the years, with a mean number of $18\pm 5$ cycles per year and a mean duration of $3 \pm1$ and $6\pm5$ days for the accumulation and melting phases of each cycle, respectively. On an annual basis, the mean number of

days with melting and accumulation dominance was 49 $\pm14$ and 108 $\pm18$ days, respectively (see page 7, lines 21-24, in the revised version).

**COMMENTS TO REFEREE #3**

**The authors analysed snow accumulation and melt cycles in four winters to determine the representative snow depletion curves which were then applied in an energy based snow accumulation and melt model. Model validation showed that the approach resulted in a reasonable reproduction of snow depth variability in an environment characterized by frequent accumulation-melt events. I am not sure that the approach which was not validated for more and larger control areas is so easily transferable to larger scales, e.g. a catchment. However, the manuscript is inspirative and I agree with its publication after modifications considering the following comments:**

First, we would like to thank Referee 3 for the comments and suggestions made to our work. All Referees in fact point out the applicability of results beyond the local scale in their remarks, and we hope to have succeeded in justifying the interest and potential of our work regarding this, as explained in the previous responses and the new text included in the revised version of the manuscript.

**- Please write what is the resolution and accuracy of snow depth values estimated from the photographs and snow poles.**

The snow depth measurements obtained by this technique are the results of a 3-step procedure: a) snow pole detection in the images, b) transformation between pixel-$h_{ref}$, and c) transformation between $h_{ref}$-$\overline{h}$. In each step, different error sources must be considered (rod resolution, number of pixel on the pole, topography, transformations, etc.). During the first year analyzed, we performed some in-situ direct measurements on one of the poles to assess this. An error of $\pm$ 30 mm was found for the $h_{ref}$ values from the comparison with these in-situ measurements. This error includes the two first sources of error. The third error source is very difficult to determine since there are no distributed snow depth measurement over the study area; to overcome this, as explained in the text, we adopted the simplifying hypothesis of a horizontal top surface of the snowpack along the cross section of the pole area, and assumed that the error stemming from this transformation is included in the resulting value of the determination coefficient ($R^2$) in the fitting step. With all, we estimated an accuracy of $\pm$ 45 mm in the snow depth measurements.

**- Equation 2 describes the relationship between href and h (for the whole control area). However, two snow poles are mentioned in data description. It is not clear if data from both poles were used in calculation of h and how.**

Only data from one pole were used in the analysis. The second one was a control test for the horizontality hypothesis adopted in the calculation of $\overline{h}$. To avoid confusion, we have rewritten this description (see p. 5, lines 25-27, in the revised version).

**- Please correct the error in the last two lines of table 2; do they not give "mean duration", but total number of days with snow accumulation and snow melt in particular winters, respectively (this corresponds to the mean values given on p. 7, l. 16).**

We apologize for this error, which has been corrected in Table 2 and the body of the text (see Table 2 and p. 7, lines 21-24, in the revised version).

**- Caption of Fig. 5 b and elsewhere in the text (e.g. page 8, line 5; p. 10., l.2) - the curve for snow accumulation (Curve 0) should not be called "depletion" curve, because the snow covered area is not decreasing, but increasing**

We fully agree with the Referee and, following this remark, we have changed the expression into ADCs, that is, accumulation/depletion curves, throughout the revised text.

**- Table 3 - please add a column with number of points (days) which were available for fitting in each cycle - Quality of Fig. 5 is not good, especially Fig. 5b is not readable**

Following this suggestion, a new column has been added to Table 3 and Figure 5 was replaced by a higher resolution version in the revised manuscript.

**- p. 9 l. 19 - Please change formulation of the sentence simulation of snow cover FRACTION should not have ME, MAE and RMSE in millimeters.**

We apologize for this, which has been corrected (see p. 10, line 3, in the revised version).

**- Fig. 8 - adding measured precipitation and air temperature to the figures would aid explanation of the mismatches mentioned in the Discussion (p. 10., l. 24-32).**

Following this, we have added the information into Figure 8 in the revised version.

**- The Discussion could also compare the results of presented approach with other works devoted to incorporation of subgrid distribution into snow models.**

Following this, we have expanded the Discussion regarding this issue (see page 12 lines 1-14 in the revised text).

**SPECIFIC COMMETS #1**

"Note to the editor and authors: As part of an introductory course to the Master programme Earth & Environment at Wageningen University, students get the assignment to review a scientific paper. Since several years, students have been reviewing papers that are in open online discussion for HESS, and they have been asked to submit their reports to the discussion in order to help the review process. While these reports are written as official reviews, they were not requested for by the editor, and we leave it up to the editor and authors to use these reports to their advantage. While several students were asked to review the same paper, this was not done to provide the authors with much extra work. We hope that these reports will positively contribute to the scientific discussion and to the quality of papers published in HESS. This report was supervised by dr. Ryan Teuling."

Summary and recommendations

To perform hydrological modelling which is GIS-based subgrid variability is important to be taken into account. Snow coverage in Mediterranean mountainous areas is highly variable. One crucial factor in the spatiotemporal distribution of snow is microtopography. Due to the heterogeneous snow distribution it is hard to calculate a snow depletion curve for semiarid regions which is representative for the whole area. Do define depletion curves snow cover fraction data over the whole study area is needed. Terrestrial photography was used in this research to derive snow depth and snow cover fraction at subgrid scales. In this study a four year series (the hydrological years 2009- 2013) of terrestrial photography images were taken in a study area in the Sierra Nevada Mountains in Spain. The images showed a high variability in snow accumulationmelting cycles. Based on these terrestrial photography images one curve was de- fined for the accumulation phase of the snow cycle and four curves were derived which are representative for the melting phases. All these four cycles are representative for different kinds of melting phases. Next a sigmoid flexible function was used to parameterize the snow depletion curves. The parameterized depletion curves were implemented in the point physical snow model described in Herrero et al. (2009) so that snow cover fraction becomes a new state variable. Simulation were performed with this model to check what the optimal model parameters were. The simulation with the best parameters shows that snow depth and snow cover fraction are simulated with a high level of accuracy. Although snow cover fraction can be overestimated in short snow accumulation-melting cycles and snow depth simulation results show different results than in reality for certain states the simulation is still accurate enough. Overall it is concluded that implementation of depletion curves that are derived from terrestrial photography improves the performance of the snow model.

In my opinion the manuscript fits perfectly to the scope of Hydrology and Earth System Sciences. The research topic is very inspiring since it is a well performed research on a small scale study area. The topic is very interesting and the research continues on paper previous research in the Sierra Nevada Mountains. They usage of depletion curves to improve the snow model is good since they are developed to characterize the decrease in snow cover fraction when snow melts (Luce and Tarboton, 2004; Rango and Martinec, 1982) and snow cover fraction is one of the important variables that is being studied in this research. Next to this I like the writing style of sections 4.2 and 4.3 very much. The results are discussed stepwise and in good detail. These sections are clearly written and they are well supported with figures and tables. The part of the discussion were the success of the depletion curves derived by the terrestrial photography is discussed is pleasant and clear to read. The

manuscript is worth publishing after some moderate revisions. I have three main comments which I will discuss in detail in the main comment section. First of all the overall message why it is so important to fulfil this research is missing. Secondly a comparison with existing literature is missing in the discussion. The conclusion that the implementation of the depletion curves in the snow model improves it performance misses comparison with quantitative results from previous studies. Next to this I also think that the methodology of the incorporation of the depletion curves in the snow model misses some explanation.

**Main comments**

The overall message why it is so important to fulfill this research is missing. In general the research is a good research which is performed with a lot of high quality techniques. These techniques are all very well integrated. However the importance of the research is not well described in the introduction. Only after reading the last paragraph of the conclusion I could make a suggestion of the importance of the study. Even though it is important to note down the importance of this research in the scientific world. By doing this the readers can be more convinced about the importance of this research. I would suggest to point out in the introduction what the advantages are of the extension of point physical snow models with depletion curves. Why is it so important to know how snow cover fraction changes on such a small scale? For example Anderton et al. (2004) tells that snow models are important in hydrological modelling and spatial variability has large influence on the prediction of surface runoff in a catchment. Another example which explains the importance of the understanding of snow cover fraction processes is that snowmelt which leads to surface runoff can only occur where the surface is covered with snow (Luce and Tarboton. 2004). I recommend to include sentences like this in the introduction and add at the end of the introduction a small paragraph where the author explains to the reader that he hopes that his results will lead to a more improved prediction of the snow cover fraction to improve the prediction of surface runoff in hydrological models.

My second main comment is about the fact that the discussion is missing comparison with previous research. The description of possible errors sources for the simulation of the model is not supported with any existing literature. It would be nice to see that the conclusions made in this paragraph are supported with previously written scientific work. Another example is that it is concluded that the results of this research are better than the results of Pimentel et al.(2015). Quantitative numbers of the accuracy of both snow model performances are not given which makes this statement hard to believe. Not only would it be better to discuss this with quantitatively results but believability of the conclusion can also increase if this research is compared to more than just one study were the performance of snow modelling is tested.

The explanation of the implementation of depletion curves in the point physical snow model is not clear enough. Right now it is impossible for other researchers to repeat this research. It is only described that the point physical is expanded by incorporating the depletion curves obtained from the terrestrial photography. However it is not explained how this is done, only that by doing this snow cover fraction will become a new state variable. Now readers can put question marks to the methodology. An unclear methodology can question the credibility of the results for the simulation of the snow model. I would suggest to explain how the depletion curves can be included in the model. Luce and Tarboton (2004) and Luce and Tarboton (1999) show that this can be done with

only one sentence; by telling that the equations for the mass balance will only be applied for the snow covered fraction. By adding a sentence like this it is clear how the implementation of the depletion curves lead to snow cover fraction as a new state variable. Next to this it was also discussed in Herrero et al. (2009) that the importance of the time step of the algorithm that solves the mass and energy fluxes is important. By including a sentence were you refer that the same algorithms as in Herrero et al. (2009) are used or an explanation why a different time step is used this problem can be solved.

First, we would like to thank Dr. Ryan Teuling for the selection of this paper as part of the introductory course of the Master Programme Earth & Environment at Wageningen University. We also thank the comments and suggestions made by the student B. Albers to our work. Following the main comments we have added some new sentences in the introduction (see page 1 lines 1-2, 7-9 and 29 in the revised text), incorporated value of the metrics obtained in previous work in the study area (see page 12 lines 1-14 in the revised text) and clarified how the new curves were implemented in the snow modelling (see page 9 lines 11-16 in the revised text) Some of the details have been also addressed in our responses to the Referees.

**Minor comments**

**Minor comment 1: In the text the five curves are most of the time all called depletion curves (for example p8, line 5), although this is not true for curve 0 since it represent the accumulation phase. Try to call only the four melting curves depletion curves and rather call curve 0 the accumulation curve.**

We fully agree with the Reviewer and, following this remark, we have split the DCs denomination into accumulation/depletion curves, ADCs, throughout the revised text.

**Minor comment 2: The terms micro-topography or microtopography are both used in the manuscript I would suggest to use only one of them. On page 11 in line 17 it is the only time that is has been written as microtopography.**

We have corrected the typo error according to this comment.

**Minor comment 3: The same as above try to stay consistent with the use of accumulation/melting cycles and accumulation-melting cycles.**

Following this, we have homogenized the denomination of the cycles choosing accumulation/melting instead of accumulation-melting.

**Minor comment 4: The reference of Luce et al. (1999) in the reference list is not published in 1999 but in 1997. This paper is used as a reference to show that in the past depletion curves have been implemented in snow physical models, but in Luce et al. (1997) depletion curves are not mentioned at all. After some research I found the paper "Sub-grid parameterization of snow distribution for an energy and mass balance snow cover model" by Luce et al (1999). This paper is published in 1999 and in this research implementation of depletion curves in snow models is discussed. I would recommend to take a look at this and include this paper in your reference list and leave out the paper by Luce et al. (1997) since it is not used anywhere else in the manuscript.**

We have corrected this typo in the reference.

**Minor comment 5: There are too many tables and figures. In the list of minor issues I have noted recommendations to combine some tables and figures. I made also a suggestion to take out one table.**

**List of minor issues**

**P1, line 30: A used reference is Luce and Tarboton (1996). In the reference list there is no reference from Luce and Tarboton from this year. So either the year number on page 1 is not correct or the reference is missing in the list.**

We have replaced the reference in the new version by the correct one Tarboton and Luce (1996).

**P2, line 20: The reference Kolbert et al. (2006) should be Kolberg et al. (2006).**

We apologize for this typo. We have corrected it.

**P3, lines 3-5: I suggest to not including this in the introduction since it is detailed methodology. The first introduction of the rods should be mentioned in the study site and available data section.**

These sentences do not aim at describing a technique; they include further usefulness of the terrestrial images, and we have included it to focus on the potential advantages with easy tools.

**P4, line 22: There is a typo in the reference Ying et al. (2003). It should be Yin et al (2003).**

We apologize for this typo. We have corrected it.

**P5, lines 6-7: This should not be included in the methodology section. The fact that 5 parameterizations are selected is a result. Therefore I suggest removing these lines from this section.**

Following this, we have removed these lines from this section.

**P5, line 19: The previous defined linear equation is not defined earlier in the text. This arises question of the credibility of your methodology section. Provide in section 3.2 also a description of this equation that is used.**

We have rewritten this sentence. We have also added some clarification along this section and a new version of Figure 2 to better understand this process (see page 5 lines 22-34 in the revised text)

**P6, lines 8 and 11: In equation 3 precipitation is defined with R, however in the text it is described that P defines the precipitation.**

We apologize for this typo. We have corrected it.

**P6, line 16: Include a reference that support why W can be disregarded. In Pimentel et al. (2016) it is stated that the rapid snow metamorphosis is observed but also in this paper a proper reference is missing. I would recommend to refer to the observations or give another proper reference.**

We have added a new reference to support this (see page 6 line 24 in the revised text)

Marks, D. and Winstral A.: Comparison of snow deposition, the snow cover energy balance, and snowmelt at two sites in a semiarid mountain basin, J. of Hydrometeorol., 2 (3), 213–227, 2001.

**P7, line 2: The year of the reference Cline (1999) mismatches the year in the reference list.**

We apologize for this typo. We have corrected it.

**P7, lines 15-16: It is stated that each year has a mean number of 18± 5 cycles a year and a mean duration of 49±14 days for the accumulation phase and 108 ±18 days for the melting phase. This tells me that one cycle has a mean duration of 49+108 = 157 days. But 18 times 157 days will exceed the amount of days in one year. This is very confusing.**

We have rewritten the sentence: The number of cycles and their duration varied considerably over the years, with a mean number of 18± 5 cycles per year and a mean duration of 3 ±1 and 6±5 days for the accumulation and melting phases of each cycle, respectively. On an annual basis, the mean number of days with melting and accumulation dominance was 49 ±14 and 108 ±18 days, respectively (see page 7 lines 21-24 in the revised text)

**Table 1: This table can be left out. I did not have the idea that this gives important information needed for the research. Rather describe more of the meteorological conditions in the text.**

This table gives a general overview about the meteorological forcing that conditions the snow results, and it helps to contextualize the Mediterranean environment to non-familiar readers.

**Table 3 and 4: Add table together since they both give information about the fitted parameters for the depletion curves.**

Tables 3 and 4 show different aspect of the fitting process, and we would rather maintain both of them.

**Table 5 and 6: Figure 6 is a very small table which continues on Table 5 since it shows the errors of the calibration parameters used in simulation 7. I would recommend to add Table 6 with Table 5.**

We have merged both tables according to this suggestion of the Reviewer (see Table 5 in the revised version)

**Figure 3: This figure is used to show that terrestrial photography can be used to determine accurate depletion curves even when the atmospheric conditions are different. Include in the caption which atmospheric conditions were present for which picture. This makes the argument more powerful.**

We have added additional information according to this comment (see Figure 3 caption in the revised version)

**Figure 5: The quality of this figure is not good. The upper part with the accumulation melting cycles is still readable, however a higher resolution would give a more neat figure. The quality of the depletion curve patterns in part b is very bad. Since I can see the shape of the accumulation curve and the depletion curves the fact that the 5 curves are very different was still clear to me. But I am not able to read the axes and titles. Due to this detailed discussion of the results cannot be followed.**

**Figure 4 and Figure 5: Since part a of Figure 5 and Figure 4 both show the accumulation-melting cycles for the calibration periods I would suggest to combine these two figures. In Figure 4 the lines of the snow depth for the cycles that are used for the calibration process can be made bold. I would recommend to add the depletion curves of part b of**

**Figure 5 underneath the calibration and validation figures.**

**Figure 6: The same accounts as for Figure 5; the depletion curves figures have a low resolution, which makes it impossible to read the figures properly.**

Following all these suggestions related to the figures, we have made some changes in Figure 6 removing the different curves and maintaining only the diagram. Nevertheless, we have decided not to joint Figures 4 and 5, since the information that they show it different in each of them.

**SPECIFIC COMMETS #2**

**Summary**

Using the combination of terrestrial photography (TP) and snow depletion curves this paper aims for a new way to describe subgrid variability of snow distribution in the Sierra Nevada. Because of the changeable climate conditions in this area, multiple accumulation and melting cycles occur throughout the snow season. Using data for snow cover fraction (SCF) and snow depth (h) from a previous four-year study (Pimentel et al., 2015), five different depletion curves (one for accumulation, four for melting) were parametrized using a flexible sigmoid function adopted from Yin et al. (2003). Subsequently, these depletion curves were implemented into a point snow model developed by Herrero et al. (2009) with the use of a decision tree. Using three years for calibration and one year for validation, simulations of the SCF and snow depth were found to be generally accurate compared to the observed values. Although the simulations in both calibration and validation periods showed an overestimation of the SCF and a mismatch in snow depth values in some states, the use of TP allowed for errors in the simulations to be related to potential error sources in measurements.

**Recommendations**

This paper is well written and structured which allows for an easy understanding of the methods used and results obtained. The introduction gives a clear overview of previous research in this field and the authors do a good job in explaining the novelty of the approach. I think the paper fits the scope of HESS particularly well because of the multidisciplinary approach, which is shown by combining the products of previous studies in order to derive new knowledge and conclusions. The use of terrestrial photography shows that cost effective methods can be integrated with a modelling approach. Although I feel this paper is almost ready for publication I think the discussion in particular needs more elaboration in order to illustrate that the methods used and results obtained are an addition to the field of snow distribution modelling.

**General Comments**

The main problem I have with this paper is that, although the approach is novel, this is not emphasized enough in the discussion. The introduction outlines flaws in the methods of previous attempts to capture subgrid variability and argues the paper's novel approach could be an improvement. The performance of the produced model was compared to the field observations to check its' general accuracy, but this is however not enough back up the claim of novelty in the introduction. The whole study is based on the expansion of the model which was developed by Herrero et al. (2009), but no comparison of the supposedly improved model is offered with the original model. There is a mention that the inclusion of the 5-curve set of DC's improved the model (page 10, line 17), this however remains quite vague while I think this should be one of the main results of the study. What would help is including a section where the new model is directly compared with the previous model by simulating the same period with both models. This quantification also allows the reader to decide for themselves if the increased accuracy is justified by the increase in effort in respect to future research.

Additionally, I think there should also be a comparison with different sets of DC's. The text mentions that the DC's are clustered if parameter values are within 10% of each other (page 5, line 4). While there is merit in clustering the DC's like this, no argumentation is given for this value. A larger value would obviously lead to less depletion curves and different simulation results. How would including only one depletion curve for accumulation and one for melting affect the model results? Increasing the number of parameters and complexity of a model will almost always improve the results but to what extend will this affect the application of this method in other areas. I think it would be useful to include a section on this in the discussion.

The introduction also mentions the problem of defining a DC for a whole watershed because of the large spatial and temporal variability in snow distribution (page 2, line 23). The authors state that using a distributed application of DC's could be used to capture this variability. However, the control area used in this study was relatively small (30mx30m) and while the conclusion proposes that this method provides a foundation for the extension of snow point models, the discussion didn't contain information about the usefulness of this method on larger scale. Could this method be applied in large areas such as a whole watershed? For example, Luce et al. (2004) defined depletion curves for the Upper Sheep Creek basin which is approximately 26 ha in size. They did this for one accumulation-melting cycle which is more common for higher latitudes, as was mentioned in the introduction. In contrast to your findings, they found little variability between years. Would this also be the case for areas such as the sierra Nevada, where multiple accumulation-melting cycles occur throughout the year. I also wonder how the terrestrial photography method could be applied in larger areas. Was the high resolution essential for the results? Would remote sensing also have worked in this situation as was done in Kolberg et al. (2006)? I think it would be useful if these upscaling issues were mentioned in the discussion. In this way, the discussion would be more in line with what the introduction stated the approach and goals of this research would be.

First, we would like to thank Dr. Ryan Teuling for the selection of this paper as part of the introductory course of Master Programme Earth & Environment at Wageningen University. We also thank the comments and suggestion made by the student E. de Badts to our work. All Referees and Reviewers in fact point out some of the points that the Reviewer mentioned in his remarks: the emphasis in the discussion comparing the results with existing literature, the explanation of how the curves are incorporated to the model, and the applicability of results beyond the local scale. These comments have been addressed in our previous answers and some changes have been introduced in the text. Please, see the different comments to the other Refeeres and Reviewers for further details.

**Specific Comments**

**I think the title should mention the use of snow depletion curves as it is a major part of this study.**

We have decided not to change the title of the paper, since our goals focus on the subgrid scale effects, and the introduction of this tool in the title could induce some misunderstanding related with its traditional use over different scales.

**The RMSE of the simulations is mentioned in the abstract but these are the RMSE values for the calibration period. I think showing the RMSE values that were found during the validation of the model would be more representative of the general accuracy.**

We have changed these values according with the Reviewer suggestion.

**Page 2, line 12: "(Mark and Dozier, 1992)" should be "(Marks and Dozier, 1992)".**

**Page 2, line 20: "Kolbert et al. (2006)" should be "Kolberg et al. (2006)".**

We apologize for these typos. We have corrected them.

**Page 4, line 15: The explanation of what exactly constitutes an accumulation melting cycle remains vague to me. Does a cycle mean the time between the accumulation of snow from a certain level and when it returns to that same level due to melting? Is it possible for multiple cycles to be ongoing at the same time? Please elaborate.**

Following this and other previous comments, we have changed the definition of accumulation/melting cycle into "…to the time period between the beginning of a snowfall event and the end of the complete ablation of the snow or the occurrence of a new snowfall event (see page 4 lines 21-22 in the revised text)

**Page 4, line 22: "Ying et al. (2003)" should be "Yin. et al. (2003)".**

We apologize for this typo. We have corrected it.

**Page 5, line 4: How did you decide upon using 10% difference for clustering the curves? There should be an argumentation for this.**

We chose a threshold for this procedure large enough to obtain more than 2 clusters. 10% was selected as standard difference or error threshold usually found in literature in many applications.

**Page 5, line 17: The explanation of how snow depth was obtained from the photos with the clustering algorithm needs more elaboration. Was it necessary to use two snow rods if the reference snow depth was used to calculate the actual average snow depth? What was the previously defined linear equation mentioned in line 19?**

**Figure 2 implies the relation between the average snow depth and the reference snow depth is linear, which it is not. The figure could be clearer on explaining the relation between the reference and average snow depth.**

We have introduced some new sentences in section 3.2. to clarify the snow depth measurement using the poles installed in the study area. We have also changed Figure 2 (see page 5 lines 22-34 in the revised text)

**Page 6, line 11: In equation 3, R is used to indicate the precipitation while the text says P.**

We apologize for this typo. We have corrected it.

**Page 7, line 21: Did excluding the cycles with short duration and cycles were the area was completely covered leave you with 16 cycles? Could use a better explanation on the method of choosing the cycles that were eventually used for calibration. It does seem from figure 5a that cycles were only used for calibration if the SCF dropped to 0. This was however not mentioned in the text.**

Yes, after the application of these two criteria we only considered 16 over 53 cycles. The cycles that the Reviewer mentioned are included in the disregarded ones.

**Table 2: In the 6th and 7th row of the table: instead of showing the mean duration of accumulation and melting cycles for the whole study period, the means of all columns are summed as was done for the 5th row.**

We have changed the denomination of these two rows: annual number of days with snow accumulation/accumulation (see Table 2 in the revised version)

**Figure 5b: The axes on the graphs of the curves are not clear.**

We have replaced them by a higher resolution version in the revised manuscript.

**Table 4: The values of h(e) in the table don't correspond with the values in the text at page 8, line 21.**

We apologize for this typo. We have corrected it, the correct one are those written in Table 4

**Table 3: I think it would be practical if the table was sorted according to the curve types**

We were thinking about this option, but we have finally decided to maintain the table in its current configuration to focus on the variability of cycles found during each analyzed year.

**SPECIFIC COMMETS #3**

This paper has derived several depletion curves and implemented them to upscale a point model to a larger grid. To do this the snow cover fraction and the snow depth are obtained using terrestrial photography using the methods from Pimentel et al.(2015). From that data and with a sigmoid function, 5 depletion curves were derived, one curve for accumulation and four curves for the melting of snow. These depletion curves were then implemented into a point snow model from Herrero et al.(2009), through a decision tree. The model was calibrated with different simulations over 3 years and after that a validation run was done for one year with the optimal simulation. The results show an accurate SCF and snow depth with similar results for both the calibration and validation. There are some errors due to weather phenomena that are not implemented into the model. The use of different depletion curves makes upscaling to larger areas possible.

I think this paper is appropriate for the journal of hydrology and earth system science because the modeling of snow is an important part of the hydrological modeling and it fits well with the scope of the paper. The research is new and innovating. The use of several depletion curves to describe accumulation and melting has never been done before and instead of the WUE like most papers, the snow depth was used for the depletion curves. This derivation of these depletion curves from terrestrial photography is an innovative approach that will be useful in further research.

This paper provides a very good research with solid methods. The methods chosen fit well together and form a consistent research together. They are well implemented from the previous literature and still very well written so that it is understandable how the methods are applied. They are to the point and explained well. Also this paper gives a very structured and good presentation of results. It is really understandable what they have done in the methods and how they came to their results. The results are given for every step of the methods, throughout the process of deriving the curves and implementing the model. This makes the process understandable and repeatable and the results credible. By chopping up the results in reasonable parts, it has a good structure which makes it nice to read. The figures and tables of the results also are understandable and are easy to understand. The introduction gives a good indication of previous methods used and it gives a good overview of history of snow modeling, making the subject more understandable The writing style of the paper is one of the lesser parts of this paper. It has been written like all the background information is already known to the reader and more details need to be implemented for better understanding. Also the discussion could be improved in several ways. Overall, I think that the research done in this paper is really well thought out and very well executed. Therefore I recommend publication for this paper with some moderate revisions following the arguments below.

First I will explain the major arguments that I think must be addressed, afterwards there are my minor comments of how to improve the paper and lastly there is a list of minor revisions.

The introduction starts with the main reason for this research. Thereafter it goes into detail of what is known and unknown and the actual goal of the research. After reading the introduction it was difficult to understand the paper since there is no context. What is missing here is an introduction of the subject of the paper, which is important for people who are no expert in the field of snow modeling. In this paper, there is only one sentence introduction(p1, line 25) and no context given. The reason for initiating this research is too short for people who have not read the background

material. It is unclear from the introduction why the scale issues are still an issue since Bloschl et al is written in 1999 and why this snow distribution modeling is important at all. This change will only affect the introduction of the paper, which can be improved in some ways. One option is a single figure in the introduction that gives the context of the subject by introducing the links between the most used terms. An explanation of the subject is also possible. In Anderton et al,(2004) and Luce et Tarboton,(2004) they first introduce the subject of snow models and explain various terms. Then they take a narrower view towards their problem that they want to solve. To broaden the reason for this research, more detail could be applied. Bloschl et al, 1999 came first with the problem of scale issues, but in this introduction there is not yet an explanation what these scale issues pertain or why they are important in the context of snow modeling so that can be implemented in the introduction.

In the discussion, the results are only compared to one other study that is a previous study using the same method, namely Pimentel et al.(2015). This is also only done in a qualitative setting by mentioning that the results are improved without mentioning what the results were of the previous study(P10, line 17-19). By not comparing the results with other papers using different methods of modeling subgrid variability, the credibility of the results can be questioned. It undermines the feasibility of this research since no quantitative comparisons are done. The addition of numbers will give a better visual presentation and makes comparison much easier for the reader. Also other papers can be mentioned that used other models or methods to derive the depletion curves such as Kolberg et al(2006), Luce et Tarboton, 2004 and also Herrero et al, 2009 (other DC's). This shows how these results compare within the field of snow distribution modeling and if the model is truly a good foundation for further upscaling of models as stated in the conclusion.

One of the goals of this paper is to give an insight in the upscaling from a point model to a 30x30 grid(P6, line 25-26). However, nowhere in the paper it is explained how this upscaling is done in the model. There is a mention that this upscaling is done by implementing the depletion curves into the model (P6, line 25-26), but by indicating that, still no explanation is given for the process of the actual upscaling. By not describing this process, an important part of the methods is left out. The research is not repeatable and less applicable for further research on upscaling. This can be remedied by explaining the process of the upscaling in the methods. This can be done for example by a step by step explanation or a short summary including the formulas used as is done in Luce et al, 1999 and Pimentel et al, 2015.

First, we would like to thank Dr. Ryan Teuling for the selection of this paper as part of the introductory course of Master Programme Earth & Environment at Wageningen University. We also thank the comments and suggestion made by the student R. Meeusen to our work. Some of the suggestions made by the Reviewer have been addressed in previous comments. Some sentences have been introduced in the Introduction to clarify different aspects (see page 2 lines 1-2, 7-8 and 23 in the revised text); moreover, specific values for the metrics obtained in previous works are added to the discussion (see page 12 lines 1-14 in the revised text); and finally some clarifications are added to the methodology section to better understand how the curves have been incorporated into the model (see page 9 lines 12-16 in the revised text).

**The minor comments:**

**Minor comment 1) The results of the RMSE mentioned in the discussion, conclusion and abstract are the results that were achieved with the calibration of the model. This seems counterintuitive since the validation of the model indicates how well the model actually works, while the calibration gives the optimal values for the model that can be achieved. Therefore the results of the validation period are more important to indicate how well the model works and should be used instead of the calibration results. The argument is given that these results are almost the same, but then please explain on what that is based, because there is still a difference in the number and if the numbers are indeed significantly the same, then still the validation results can be used for the comparison to other papers.**

We have revised and changed these values in the abstract according to the Reviewer suggestion (see page 1 line 22 in the revised text).

**Minor comment 2) The second paragraph of the discussion (line 8-11) is unclear in what is meant. Figure 9 is a important figure but does not seem understandable now. An explanation is needed of what is exactly visible in the different pictures, and what is different between the pictures with same SCF. In line 8-11 an explanation of weather phenomena are given that are not clearly visible in figure 9. Please rewrite this paragraph and give an explanation of the differences visible, and in weather conditions, in the pictures in figure 9.**

**Minor comment 3) The third paragraph of the discussion (line 12-16) is unclear in their goal. The arguments given in this paragraph do not seem to relate to the conclusion in the last sentence of this paragraph. Please explain this relation better and why that conclusion can be drawn. Also that conclusion relates to the last sentence in this paper, which is a conclusion that this research provides a basis for extension of snow models to larger areas. However these conclusions do not say the same thing. Please relate the two conclusions better with each other and give argument for these conclusions. Also explain why these conclusions can be drawn from the results because that is now not completely clear in the paper.**

We have added more discussion of the results following these two Minors comments 2 and 3 (see page 10 lines 19-34 and page 11, lines 1-14 and lines 24-34 in the revised text)

**Minor comment 4) This paper has a good explanation of DCs. The description of the method of acquiring these depletion curves is very accurately described and it is well thought out how they would use the data to derive the depletion curves. Especially page 8 gives a very clear overview of each depletion curve that is used. This part is very nice and well written.**

**Minor comment 5) In paragraph 3.2 the methods for acquiring the snow depth with the rods is explained. However is unclear to me how the reference snow depth is calculated. Figure 2 does not help in explaining this method. Please give more explanation of the method and explain figure 2 in more detail. Also why the different snow depths are visible at the levels that they are located on.**

We have introduced some new sentences in section 3.2. to clarify the snow depth measurement using the poles installed in the study area. We have also changed Figure 2 (see page 5 lines 22-34 in the revised text).

**Minor comment 6) At the very end of section 4.2 on page 8 there is said that a decision tree is implemented into the model, but it is not described how this is done. The decision tree is not even mentioned in the methods. Please describe in the methods how this decision tree is implemented in the model.**

Following this, we have introduced some new sentences explaining how the decision tree is used by the model (see page 9 lines 11-16 in the revised text).

**Minor comment 7) At the end of page 10, the error sources of the depletion curves are given. Both the first and third error also give a possible explanation for these errors. However the second error does not have any explanation of how this insufficiency of rain-over-snow effects is caused. Please indicate the reason for this error, or indicate that it is unclear how this error is caused.**

The model does not include a specific formulation to capture this effect. It could be a future improvement on the model.

**Minor revisions:**

**P1, line 17-18: "The resulting DCs were able to capture certain physical features of the snow, which were used. . ." seems like the physical features were used and included into the model instead of the depletion curves, so the sentence structure could be changed.**

**P1, line 29-30:" Luce et Tarboton, 1996" is not mentioned as such in the references, should be Tarboton et Luce, 1996 or reference needs to be changed.**

**P2, line 20: "Korbert" should be "Kolberg".**

We apologize for all these typos. We have corrected them in the revised text.

**P2, line 28-31: These sentences can be moved to the methods, since they describe the possible shapes for depletion curves.**

These sentences describe the  general pattern that accumulation/melting cycles follow, rather than the curves used to quantify their evolution, and that is why this is included in the Introduction.

**P3, line 3-5: These sentences can be moved to the methods, since this information is missing there, and if it all information on the methods of the rods is put together, the explanation of the rods is more understandable.**

These sentences do not aim at describing a technique; they include further usefulness of the terrestrial images, and we have included it to focus on the potential advantages with easy tools.

**P4, line 22:"Ying et al" should be "Yin et al".**

We apologize for this typo. We have corrected it.

**P5, line 6-7: This sentence pertains results and does not have to be mentioned in the methods.**

Following this, we have removed these lines from this section.

**P5, line 19: "A previous defined function" has not been defined in this paper, so please put it in the paper.**

We have rewritten this sentence to avoid confusion (see page 5, line 27, in the revised version and new Figure 2).

**P6, line 11: "P" is not mentioned in formula, should be "R" that is mentioned in formula above, or the "R" should be changed into a "P".**

We apologize for this typo. We have corrected it.

**P7, line 15-16: The paper mentions 18 cycles per year with a duration of 49+108=157 days for each cycle. This seems very illogical since it seems that the cycle duration is too long to fit 18 times in one year. Either a bit more explanation that cycles can overlap is necessary or the sentence structure needs to be changed.**

We have rewritten the sentence: The number of cycles and their duration varied considerably over the years, with a mean number of $18\pm 5$ cycles per year and a mean duration of $3\pm 1$ and $6\pm 5$ days for the accumulation and melting phases of each cycle, respectively. On an annual basis, the mean number of days with melting and accumulation dominance was $49\pm 14$ and $108\pm 18$ days, respectively (see page 7, lines 21-24, in the revised version).

**P7, line 26-28: These sentences can be moved to the methods. The amount of detail given here belongs in the methods, not in the results.**

**P8, line 28-29: This sentence belongs in the methods, since it is not mentioned there.**

**P9, line 2-4: These sentences belong in the methods. The decision tree is not mentioned in the methods and this amount of detail should be mentioned in the methods.**

In fact, these sentences focus on aspects that were derived after the obtaining of some results or during their analysis, and anticipating them in the methods make it complex to maintain a clear order of reasoning. We have maintained the current version due to this.

**P11, line 3: it says "error sources of error" so one error can be left out.**

We apologize for this typo. We have corrected it.

**P11, line 28: It says "error of less than", but in the rest of the paper these amounts are given as the error, not a smaller value as is insinuated here.**

We apologize for this typo. We have corrected it (see page 13 line 27 in the revised text).

**P14, line 5-6: I think the wrong reference title is mentioned here, since the paper with that title is from 1997, and does not mention depletion curves, while the paper that is referred too does mention DCs as indicated in the text P2 line 14.**

We apologize for this typo. We have corrected it.

**P16, table 1: This table can be left out of the paper, since this figure does not have added value to the understandability of the paper.**

**P17, table 2: This table can be left out of the paper, since this information is not necessary to understand the paper, and is not explained in the text.**

Table 2 sums up the changeable characteristics of accumulation/melting cycle on semiarid areas. We have maintained it.

**P 20/21, table 5 & 6: these figures can be combined into one figure since table 6 is very small.**

Following this, we have combined both tables in a new Table 5.

**P 23, fig 2: it is unclear how href and h1 and h2 came to be in this figure, more explanation in the header can give more understanding of what is meant with each parameter visible in the figure.**

We have introduced some clarification in section 3.2 and in Figure 2 (see page 5 lines 22-34 in the revised text)

**P24, fig 3: The header of the figure could use an explanation about the differences in weather condition between the three different dates, as it is now quite unclear what makes them different since now especially the first and last date seem to indicate the same conditions.**

We have introduced more information in the caption of Figure 3 following the suggestion of the Reviewer.

**P 25/26 fig 4 & 5: both figures can be combined, the only difference in figure 4 with 5 is that all cycles are mentioned in figure 4 instead of only the cycles used in this paper as in figure 5. Placing two of the same figures in the paper with little difference in information is unnecessary.**

We have decided not to joint Figures 4 and 5, since the information that they show it different in each of them.

**P26, fig 5: axis of the DC curves are not readable.**

**P27, fig 6: axis of the DC curves are not readable. P27, fig 6 "More than 30 days with previous snow" is unclear what is meant, so it would be good to reformulate.**

We have improved the resolution quality of both Figures.

[revised manuscript text omitted]